# A GPU-accelerated Large-scale Simulator for Transportation System Optimization Benchmarking

## Abstract

With the development of artificial intelligence techniques, transportation system optimization is evolving from traditional methods relying on expert experience to simulation and learning-based decision and optimization methods. Learning-based optimization methods require extensive interactions with highly realistic microscopic traffic simulators. However, existing microscopic traffic simulators are inefficient in large-scale scenarios and thus fail to support the adoption of these methods in large-scale transportation system optimization scenarios. In addition, the optimization scenarios supported by existing simulators are limited, mainly focusing on the traffic signal control. To address these challenges, we propose the first open-source GPU-accelerated large-scale microscopic simulator for transportation system simulation and optimization. The simulator can iterate at 84.09Hz, which achieves 88.92 times computational acceleration in the large-scale scenario with 2,464,950 vehicles compared to the best baseline CityFlow. Besides, it achieves a more realistic average road speeds simulated on real datasets by adopting the IDM model as the car-following model and the randomized MOBIL model as the lane-changing model. Based on it, we implement a set of microscopic and macroscopic controllable objects and metrics provided by Python API to support typical transportation system optimization scenarios including traffic signal control, dynamic lane assignment within junctions, tidal lane control, congestion pricing, road planning, e.t.c. We choose five representative transportation system optimization scenarios and benchmark classical rule-based algorithms, reinforcement learning algorithms, and black-box optimization algorithms in four cities. These experiments effectively demonstrate the usability of the simulator for large-scale traffic system optimization. The anonymous code of the simulator is available at `https://anonymous.4open.science/r/moss-AF45` and the others are shown at Appendix A. In addition, we build an open-registration web platform to support no-code trials.

## 1 Introduction

With the increasing level of urbanization and residents' travel demand, the urban transportation system faces heavier traffic pressure, which brings higher commuting costs, environmental pollution and other society problems, affecting the sustainable development of the city (Kirago et al., 2022; Wang et al., 2021; Treiber et al., 2008). To alleviate the above problems, governments usually build more transportation infrastructure and optimize the existing transportation infrastructure to enhance the systems' capacity. For instance, these transportation system optimization methods include traffic signal control, congestion pricing, etc. However, the traditional transportation system optimization process is highly dependent on the experience of experts, which is labor-heavy and often sub-optimal (McNally, 2007). With the development of reinforcement learning (Mnih et al., 2013; Schulman et al., 2017) and black-box optimization (Hansen et al., 2010; Costa & Nannicini, 2018), the above optimization methods have great potential for improving the transportation system. But since all of these optimization methods use extensive interactions with the environment for feedback to perform optimization, it requires that the environment be able to model the transportation system as realistically as possible and can provide feedback fast. In the field of transportation system, simulators

that use kinetic formulations with individual motion models to calculate individual motions to provide realistic results are referred to as microscopic simulators.

At present, there are several available microscopic simulators that can evaluate the efficiency of the transportation system, including SUMO (Behrisch et al., 2011), CityFlow (Zhang et al., 2019), and CBLab (Liang et al., 2023). However, these simulators face the following two key challenges:

- **Computational inefficiency in large-scale scenarios.** Since the urban transportation system is a complex system with strong direct spatial and temporal correlation between different regions, the traffic improvement in a one area may lead to congestion in another area. Therefore, the effect of transportation system optimization should be evaluated in a global city-level perspective, which poses a requirement for large-scale microscopic simulation. However, existing simulators typically use CPUs for computation, making it difficult to simulate city-level scenarios with a large number of individuals fast. For example, the most popular open-source simulator, SUMO, is still using a single-threaded computing architecture, which significantly reduces the efficiency of the data sampling process of optimization algorithms such as reinforcement learning. Even though CityFlow and CBLab use multi-threading techniques, it still takes more than 100 seconds to simulate 1 hour in a scenario of about 100,000 vehicles. Due to the large number of environmental interactions required by optimization methods, especially learning-based ones, existing simulators fail to support the adoption of these methods in large-scale transportation system optimization scenarios. Therefore, we need a simulator capable of simulating large-scale scenarios with about 1 million vehicles at a frequency of at least 10 Hz for supporting it.

- **Limited supported optimization scenarios.** In order to improve the efficiency of the transportation system, traffic management authorities usually apply a variety of transportation system optimization methods, including traffic signal control optimization, intersection lane turn assignment, tidal lane, congestion pricing, etc. If these methods can be used jointly, transportation system efficiency improvements will be further enhanced. However, existing simulators and related optimization studies usually focus on only a few of these scenarios, such as the traffic signal control optimization problem (Wei et al., 2019; Wu et al., 2023b), ignoring other scenarios. This situation prevents traffic managers from fully evaluating and comparing the effectiveness of various transportation system optimization methods from microscopic like traffic signal control (Zheng et al., 2019; Wei et al., 2019; Wu et al., 2021; Zhang et al., 2022) to macroscopic like congestion pricing (Buini et al., 2018; Pandey & Boyles, 2018). To improve it, the simulator should be implemented to be able to support more common transportation system optimization methods and scenarios.

To address the above challenges, considering the characteristics of individual independent computation in microscopic simulation matches the GPU architecture and the massive computational power of GPUs compared to CPUs, we propose the first open-source GPU-accelerated large-scale microscopic simulator. This simulator adopts a parallel-friendly design of computational flow and data partitioning, and designs an efficient indexes for sensing between vehicles. Based on these, we implement microscopic traffic simulation on CUDA and substantially improves the scale and efficiency of simulation. In the largest scenario with 2,464,950 vehicles, this simulator can iterate at 84.09Hz, which is 88.92 times better than the optimal baseline CityFlow. it achieves a more realistic average road speeds than CityFlow simulated on real datasets by adopting the IDM model as the car-following model and the randomized MOBIL model as the lane-changing model. To support the optimization of various scenarios, it also implements a set of microscopic and macroscopic controllable objects and metrics, and provides a Python application programming interface (API). By combining controllable objects and metrics, we implement five typical transportation system optimization scenarios including traffic signal control, dynamic lane assignment within junctions, tidal lane control, congestion pricing, and road planning for benchmarking and evaluate the performance of classical rule-based algorithms, reinforcement learning algorithms and black-box optimization algorithms for these scenarios in 4 large cities including Beijing, Shanghai, Paris, and New York. The experiments show the usability of the simulator for large-scale traffic system optimization.

In short, our contribution are two-fold. First, we propose a realistic high-performance large-scale microscopic simulator for transportation system simulation on GPU and implement microscopic and macroscopic controllable objects and metrics to support transportation system optimization. Second, we choose and implement five typical transportation system optimization scenarios and benchmark common optimization algorithms in four cities to show the usability of our proposed simulator.

Table 1: Comparison of microscopic simulators for transportation system. The *Scale* field indicates the approximate number of vehicles that can be computed by the simulator at a simulation computation frequency of 10Hz on an Intel(R) Xeon(R) Platinum 8462Y CPU (64 threads) and an NVIDIA GeForce RTX 4090 GPU. (Citations are left in the text to avoid out-of-width.)

| | Simulator | SUMO | CityFlow | CBLab | Ours |
|---|---|---|---|---|---|
| | Scale (10Hz) | $< 10,000$ | $\sim 130,000$ | $\sim 150,000$ | $> 10,000,000$ |
| Simulation Models | Car-following Lane-changing | Selectable[1] | Krauß Earliest | Krauß Private | IDM Randomized MOBIL |
| Controllable Objects | Traffic Signal | ✓ | ✓ | ✓ | ✓ |
| | Lane/Road Max Speed | ✓ | ✗ | ✓ | ✓ |
| | Lane Function | ✗ | ✗ | ✗ | ✓ |
| | Vehicle Route | ✓ | ✓ | ✓ | ✓ |
| Metrics | Lane Queue Length | ✓ | ✓ | ✓ | ✓ |
| | Road Travelling Time | ✓ | ✗ | ✗ | ✓ |
| | Average Travelling Time | ✓ | ✓ | ✓ | ✓ |
| | Throughput | ✓ | ✗ | ✗ | ✓ |

# 2 RELATED WORKS

## 2.1 EXISTING SIMULATORS FOR TRANSPORTATION SYSTEM

Existing simulators for transportation system can be divided into three categories based on the level of simplification of the simulation models: microscopic simulators, mesoscopic simulators, and macroscopic simulators. Macroscopic simulators (Mahmassani, 1992; Group) typically do not consider modeling individual vehicles, but rather treat the vehicles as a fluid for using velocity and density to describe them. Mesoscopic simulators often speed up the simulation by simplifying the vehicle motion models. For instance, MATSIM (W Axhausen et al., 2016) use a uniform motion model with intersection waiting queues (Gawron, 1998) to model vehicles and do not consider acceleration and deceleration. Since macroscopic and mesoscopic simulators oversimplify vehicle motion, they are not usually used for AI algorithm based transportation system optimization. Among the microscopic simulators, SUMO (Behrisch et al., 2011), CityFlow (Zhang et al., 2019), and CBLab (Liang et al., 2023) shown in Table 1 are popular simulators for transportation system optimization. They all use a car-following model and a lane-change model to simulate the vehicle's behaviors and calculate the acceleration, velocity, and position of vehicles. SUMO provides multiple simulation models for user selection and offer a rich set of controllable objects and metrics. However, due to its software architecture, SUMO can almost exclusively use one CPU core for computation, which leads to small simulation scales. For CityFlow and CBLab, they both use a multi-threaded architecture for computational acceleration, which improve computational speed by about $20 \sim 30$ times on 64-threaded CPUs relative to SUMO. But with city-scale simulations of at least 100,000 vehicles, it still takes minutes for them to simulate an hour, which constrains the speed of reinforcement learning algorithms to learn by interacting with the environment. Both CityFlow and CBLab use Krauß et al. (1997) as the car-following model. For the lane-changing model, CityFlow uses an as-early-as-possible lane changing strategy, while CBLab implements its own private adaptive lane changing algorithm. Besides, in terms of controllable objects, CityFlow only provides interfaces for setting traffic signal phases and vehicle routes while CBLab adds the setting of road speed limits as an additional feature. In terms of metrics, both CityFlow and CBLab provides lane queue length and average traveling time (ATT) directly. Most of these controllable objects and metrics are designed for traffic signal optimization, and other optimization scenarios cannot be directly implemented accordingly. Overall, there is a lack of microscopic simulators that can effectively simulate and provide rich controllable objects and metrics to support transportation system optimization problems in large scale scenarios.

---

[1]Shown in the *Car-Following Models* and *Lane-Changing Models* sections at `https://sumo.dlr.de/docs/Definition_of_Vehicles%2C_Vehicle_Types%2C_and_Routes.html`

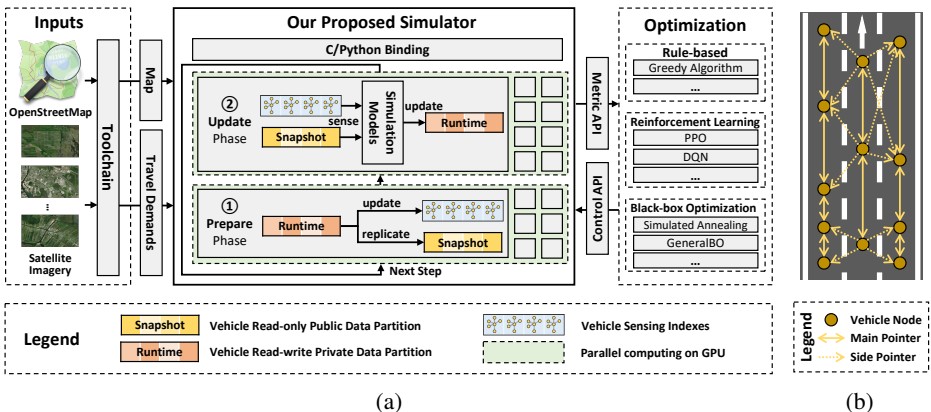

Figure 1: (a) The framework of the proposed simulator. (b) The linked-list of the center line for vehicle sensing by one pointer operation. (best view in color)

## 2.2 EXISTING TRANSPORTATION SYSTEM OPTIMIZATION METHODS

Existing methods for optimizing transportation systems can be classified into rule-based and learning-based methods. Rule-based methods use expert experience to design and improve rules, relying on rules for control and optimization, e.g. the maximum pressure algorithm (Varaiya, 2013) in traffic signal control and the $\Delta$-*tolling* algorithm (Sharon et al., 2017) in congestion pricing. Such methods are difficult to adapt to complex and changing traffic conditions and only consider local optimization. Learning-based methods usually use reinforcement learning (Mnih et al., 2013; Schulman et al., 2017) to find the global optimal solution by making a large number of tries in a simulation environment. The traffic signal control problem is the most extensively studied problem in the field of transportation system optimization, with both rule-based methods (Varaiya, 2013) and learning-based methods (Zheng et al., 2019; Wei et al., 2019; Wu et al., 2021; Zhang et al., 2022; Wu et al., 2023a; Oroojlooy et al., 2020). In the congestion pricing problem, the reinforcement learning algorithm has also been adopted (Buini et al., 2018; Chen et al., 2018; Pandey & Boyles, 2018; Qiu et al., 2019; Wang et al., 2022). Comparatively, other transportation system optimization scenarios such as dynamic lane assignment (Li et al., 2009; Zhou et al., 2019; Jiang et al., 2021), tidal lane control (Li et al., 2013; Zhang & Tang, 2021; Li et al., 2023), etc. do not seem to have received much attention from researchers. And existing works only focus on small-scale problems. This is most likely due to the lack of simulators that support multiple scenarios including those mentioned above.

## 3 THE SIMULATOR

To address two key challenges in the simulation and optimization of large-scale urban transportation systems, we first introduce to the design of our proposed simulator for efficient microscopic traffic simulation and then discuss the microscopic and macroscopic controllable objects and metrics with their APIs to support optimization in the section. The framework of the proposed simulator is shown in Figure 1(a).

### 3.1 SYSTEM DESIGN

Microscopic traffic simulation is the process of modeling and discrete-time simulation calculations for each vehicle in the transportation system. Performing one step simulation usually represents simulating a 1-second change in the real world. When facing large-scale scenarios with hundreds of thousands of vehicles, the large number of vehicle model calculations will consume a lot of computational power, resulting in a low running speed.

The development of modern computational acceleration hardware provides the basis for a solution to this problem. Single instruction multiple data (SIMD), as the basic computational model of hardware acceleration cards such as GPUs, trades off instruction flexibility for the ability to parallelize a large

number of homogeneous tasks and has been used with great success in areas such as matrix arithmetic acceleration and 3D image rendering. In microscopic traffic simulation, the simulation models of individual vehicles are also highly homogeneous and therefore highly compatible with the SIMD computational model.

However, before we can implement vehicle simulation models as CUDA code, we need to address the two problems posed by the need for vehicles to sense each other. First, in an iteration, the vehicle needs to read the position, velocity, and other attributes of other vehicles as inputs to the simulation models for computing appropriate driving behaviors such as accelerating, decelerating, and changing lanes. Thereafter, the vehicle will also modify its own attributes such as position, velocity, etc. based on the driving behavior of the decision. This leads to **the problem of read/write conflict of vehicle data**, which will affect the correctness of the simulation results. Second, the range that the vehicle senses includes only the front vehicle in the current lane and the front and rear vehicles in the adjacent lanes, which is spatially localized. Thus, implementing **SIMD-friendly vehicle sensing indexes** for the above retrieval task is the key to fully utilize the massive arithmetic power of the modern computational acceleration cards. The two key designs of our proposed simulator to solve the above two problems respectively are described below.

**Two-phase Parallel Process for Read/Write Separation.** In order to ensure that vehicles always correctly read the previous step's attributes of other vehicles and to avoid the read/write conflict, we divide the vehicle's attributes into two partitions: *snapshot* and *runtime*. The *snapshot* is a read-only data partition that always saves the public attributes of the previous step for other vehicles to access. The *runtime* is a private and read-write partition, the attributes of which are changed after the vehicle completes its simulation calculations. In order to implement the data replication from the *runtime* partition to the *snapshot* partition, we also divide each iteration into two sequential phases, the *prepare* phase and the *update* phase. The *prepare* phase is used to perform vehicle data replication in parallel and update the vehicle sensing indices based on the new snapshot data. In the *update* phase, the vehicle performs sensing to obtain the attributes of the *snapshot* partitions of other vehicles and performs the IDM model (Treiber et al., 2000) as the car-following model and the randomized MOBIL model (Kesting et al., 2007; Feng et al., 2021) as the lane-changing model to update its own *runtime* partition attributes. The above process effectively avoids the read/write conflict of vehicle data, which on one hand ensures the correctness of the calculation results, and on the other hand makes the calculation flow more suitable for the SIMD calculation model due to the mutex-free structure and the highly homogeneous calculation procedures.

**Linked-list based Vehicle Sensing Indexes.** The key to design a SIMD-friendly spatial relative position indexing is to pick the suitable data structure. Since branching will lead to the thread divergence and reduce the execution efficiency under SIMD architecture, the adjacent element query operation with constant time complexity is required. Because vehicle lane changes will result in random insertions and deletions, such actions of the data structure should be efficient. Commonly used ordered structures include ordered vectors, ordered linked lists, and binary trees. Binary trees are discarded due to their inconstant adjacent element query. The insertion and deletion of ordered vectors will result in a large number of data movement operations, resulting in on-chip memory bandwidth pressure. Therefore, ordered linked lists are the most suitable candidates as the vehicle sensing indexes. In order to solve the query with constant time complexity for the front and rear vehicles in the adjacent lanes, we add side pointer design to the bidirectional ordered linked list. Shown in Figure 1(b), one linked list records all vehicles in order of spatial location in one lane. Each vehicle node has two pointers to the front and rear vehicle in the same lane respectively, named as main pointers. In addition, we add four pointers pointing to the front and rear vehicles in the left and right lanes respectively, named as side pointers. With such an index structure, vehicle sensing always requires only one pointer operation, avoiding the thread divergence problem. Since there are usually only a small number of vehicles entering or leaving the lane at each iteration, the number of operations including adding nodes, deleting nodes, and reordering of the linked list during the index update process is relatively small so that the impact on the computational performance is acceptable. With this design, we address the second problem by providing a SIMD-friendly vehicle sensing indexes with low update cost for vehicle simulation model computation.

To make it user-friendly, we provide its Python API and a toolchain for building its inputs.

**Python API.** The simulator exposes the C interfaces as Python API. The Python API consists of a series of initialization functions, getter functions, setter functions, and the `next_step` function

that control the progress of the simulation. The setter functions usually provide batch versions additionally with the `_batch` suffix to minimize the data transfer overhead for large numbers of calls. We will mention some of the key APIs in Section 3.2 and Section 3.3 and leave most of the general APIs like `get_vehicle_speeds()` in the public documentation. This Python API dose not directly provide the gymnasium-style reinforcement learning environments, but rather requires users to build the environment by combining the above functions according to the need of scenarios. The benchmark open-source codes for the five scenarios provides examples of how to encapsulate the gymnasium-style reinforcement learning environments on top of the Python API.

**Simulator Inputs and Toolchain.** Following microscopic traffic simulation setup, the simulator inputs are map data and travel demand. The map data describes the geospatial attributes and topological relationships of road networks and the candidate traffic signal phases of junctions. Travel demand describes the vehicle's origin, destination, departure time, and chosen route. These inputs are stored in a binary format defined by Protobuf[2]. In order to facilitate the construction of simulator inputs, we have developed a open-source toolchain. The toolchain mainly provide map building based on OpenStreetMap[3] and real travel demand generation based on globally available public data represented by satellite imagery. By using the toolchain, users can easily build maps, generate travel demands, and subsequently begin simulation and optimization.

Appendix B will introduce the system implementation details, including the execution process, vehicle model implementation, etc., and further compare the system with baselines in terms of system implementation for the interested readers.

## 3.2 CONTROLLABLE OBJECTS

To support the major transportation system optimization scenarios, we set up the following APIs for the controllable objects of the transportation infrastructure and traffic participants, where the simulator instance in Python is always labeled with `engine`.

**Traffic Signal.** The simulator allows the user to set the traffic signal control policies for given junctions via `engine.set_tl_policy(id, policy)`. The policy enumeration includes MANUAL, FIXED_TIME, MAX_PRESSURE and NONE. Under the MANUAL policy, the user can change the current phase and duration of the signal via `engine.set_tl_phase(id, phase_index)` and `engine.set_tl_duration(id, duration)`. The FIXED_TIME policy indicates that the fixed phase procedure built into the map data is used. The MAX_PRESSURE policy indicates that the adaptive maximum pressure algorithm (Varaiya, 2013) is used. The NONE policy indicates that there is no signaling.

**Lane.** Lanes in the simulator include both clearly marked lanes on the roadway and "virtual" lanes within junctions that connect the two roadways. For lanes, the user can first set their maximum speed via `engine.set_lane_max_speed(id, max_speed)`. Secondly, the user can set whether the lane is restricted from passing via `engine.set_lane_restriction(id, flag)`.

**Road.** To support dynamic changes in lane function combinations, the roadway is pre-configured with multiple lane function combination plans. Of these, lane functions are referred to as being used for going straight, turning left, and turning right. The user can set the road's lane function plan via `engine.set_road_lane_plan(id, plan_index)`.

**Vehicle.** The user can change the route of the vehicle via `engine.set_vehicle_route(vehicle_id, route, end_lane, end_s)` to modify its route and destination.

In addition to these controllable objects, the user can also change the map before simulating to build optimization scenarios.

## 3.3 METRICS

To make it easier for users to calculate common microscopic and macroscopic metrics, we also provide the following metric APIs.

---

[2]`https://protobuf.dev/`
[3]`https://openstreetmap.org/`

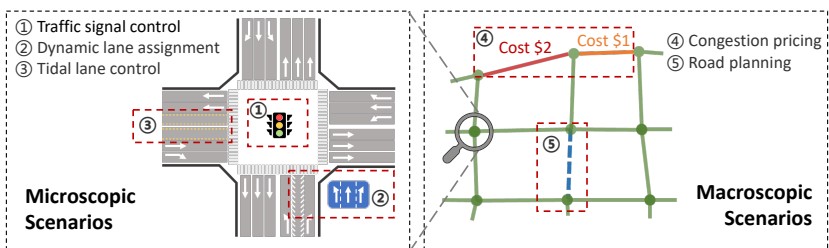

Figure 2: The overview of the five transportation system optimization scenarios. (best view in color)

**Lane Queue Length.** Lane queue length is used to count the number of vehicles waiting to be released at the end of the lane, which is a microscopic metric often used as an input to traffic signal control algorithms. The metric is provided via `engine.get_lane_waiting_at_end_vehicle_counts()`.

**Road Traveling Time.** Road traveling time indicates the time taken by vehicles to pass through the road under the current traffic flow on the road, which is a microscopic metric that directly shows how congested the road is. The metric is provided via `engine.get_road_average_vehicle_speed()`.

**Average Traveling Time.** Average traveling time (ATT) is the average time taken by all vehicles to complete a trip. It is a commonly used macroscopic metric that directly reflects the overall efficiency of the transportation system. The metric is provided via `engine.get_finished_vehicle_average_traveling_time()`.

**Throughput.** Throughput (TP) is used to indicate how many vehicles complete a trip in a given time period. It is also commonly used as a macroscopic metric for assessing the efficiency and capacity of a transportation system. The metric is provided via `engine.get_finished_vehicle_count()`.

## 4 TRANSPORTATION SYSTEM OPTIMIZATION SCENARIOS

As shown in Figure 2, we choose three microscopic optimization scenarios and two macroscopic ones for benchmarking. The former ones focus on both junction-level and roadway-level transportation infrastructure control. The latter ones include pre-construction planning phase as well as the post-construction management phase.

**Traffic Signal Control.** Traffic signal control is the most convenient approach to optimize the transportation system, which is also the scenario where the AI optimization methods are most widely used in the research field of transportation. The approach adjusts the phase and duration of traffic signals at junctions to control the number of vehicles passing in different directions, making full use of road resources to reduce the time spent by vehicles in the transportation system. Therefore, the appropriate setting of signal phasing and timing taking into account the interactions between junctions will substantially affect the efficiency of the transportation system.

**Dynamic Lane Assignment within Junctions.** Dynamic lane assignment within junctions refers to the adaptive reallocation of lane functions, such as for straight, left turn or right turn, across all lanes at the junctions based on real-time traffic conditions. For example, when there is an increase in the number of left-turning vehicles in a particular direction at an junction, the method will increase the number of lanes on the corresponding roadway used for left-turning and decrease the number of lanes used for going straight, thereby decreasing the waiting time for vehicles at the junction. How to make the correct dynamic lane assignment based on the current situation and the prediction of the future is an important transportation system optimization problem.

**Tidal Lane Control.** Tidal lanes are a classical traffic management strategy to manage the increased traffic pressure that is predominantly in one direction during morning and evening rush hours. This method increases roadway capacity and reduces congestion by redirecting lane usage. For instance, during the morning rush hour, more lanes might be designated for inbound traffic, while in the

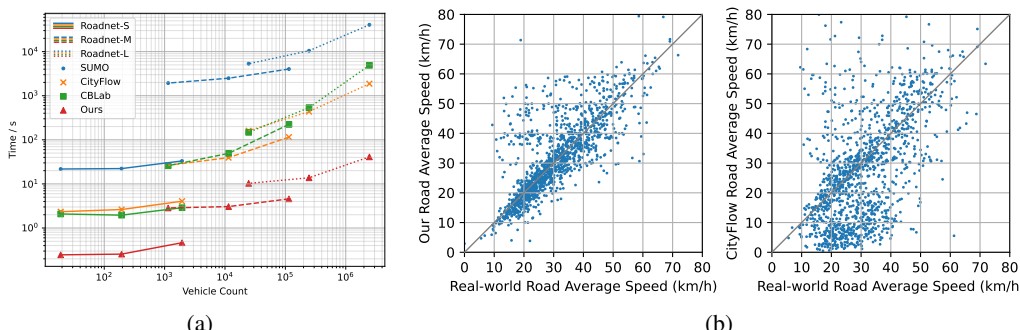

Figure 3: (a) The performance comparison with different sizes of road networks and number of vehicles. (b) Comparison of real-world and simulated average vehicle speeds. (best view in color)

evening, the direction is reversed to accommodate outbound traffic. Thus, optimization of the timing and direction of tidal lane adjustments can improve commuting efficiency throughout the city.

**Congestion Pricing.** Congestion pricing is a macroscopic traffic management strategy that uses congestion charges for vehicles driving into specific areas or roads to control and reduce traffic flow, thereby improving traffic conditions. Through such pricing tactics, vehicles will change to routes with lower costs. From a global perspective, a good pricing strategy will balance the traffic flow and traffic pressure in different areas, and thus improve the overall traffic congestion situation.

**Road Planning.** Building new roads is the most direct way to increase the carrying capacity of the transportation system. Properly planning the location of new roads and their relationship to existing roads is a prerequisite for maximizing the return on investment. In this scenario, we consider a numerous set of potential new road candidates and use optimization approaches to identify the road combinations that are optimal in terms of efficiency improvement of the overall transportation system under specific constraints such as total distances, total investment, number, etc.

## 5 EXPERIMENTS

**Simulator Performance.** To illustrate the computational performance of our proposed simulators, we compared the computational efficiency of simulators including SUMO, CityFlow, CBLab, and our proposed simulator for different road network scales and vehicle sizes. We adopt the datasets from CBLab (Liang et al., 2023) and choose three road networks of different sizes. For each road networks, we choose three travel demands of different sizes. We simulated 3600 steps for all the datasets and record the total running times as the performance of each simulator. As shown in Figure 3(a), the result indicates that our proposed simulator has a huge performance improvement over existing simulators under all conditions. On the largest dataset, the running time of ours is 42.81s and that of the best baseline (CityFlow) is 3806.7s, a relative performance improvement of 88.09 times. The statistics of the datasets and hardware used are left in Appendix C.

**Simulator Realism.** To indicate its realism, we compared the similarity between our and CityFlow's (the best baseline in performance comparison) simulation results and the real traffic situation dataset from Shenzhen, China (Yu et al., 2023). The dataset contains 4.22 million vehicle GPS trajectories as the ground truth of average vehicle speeds and 156,856 vehicle trips captured by traffic cameras as the input. The experiment compares the simulated average vehicle speeds of 1,341 roads from 8 to 9 AM and the results are shown in Figure 3(b). The average road speeds obtained from the our simulator (RMSE=8.5km/h, correlation coefficient=0.7691) are closer to the real-world data compared to CityFlow (RMSE=16km/h, correlation coefficient=0.5286), which shows that our simulator achieves a more realistic simulation by adopting the newer car-following model (IDM) and lane-change model (randomized MOBIL). The visualization results of road traffic status are shown in Figure A3.

To benchmark the optimization algorithms for the five scenarios described above, we chose Beijing, Shanghai, Paris and New York as test cities. The road networks of these cities are built using the toolchain. The real origin-destination (OD) matrices of these cities are also generated by the toolchain

using generative AI methods. As synthetic datasets, in terms of vehicle departure times, we kept only the morning and evening peaks to challenge the optimization algorithms for each scenario. The total number of vehicles was scaled based on the generated real OD matrix to construct travel demand data for three different congestion levels including smooth (marked as City-S), normal (marked as City-N), congested (marked as City-C). More information on the synthetic datasets will be provided in Appendix E. We evaluate the optimization effectiveness of different algorithms under the above cities and congestion levels, using ATT and TP as global metrics for comparison. In the following text, the comparisons of the various optimization algorithms used in all the five scenarios and their performance under normal congestion will be reported. The detailed experimental settings and the complete results are presented in Appendix F and Appendix G, respectively.

**Traffic Signal Control Benchmark.** In this scenario, the task is to choose the best traffic signal signal phase from the list of available phases for each junction. We compared the rule-based algorithms, including fixed-time algorithm (Koonce & Rodegerdts, 2008) and maximum pressure algorithm (Varaiya, 2013), and the reinforcement learning-based algorithms, including FRAP (Zheng et al., 2019), MPLight (Chen et al., 2020), CoLight (Wei et al., 2019), Efficient-MPLight (Wu et al., 2021), Advanced-MPLight and Advanced-Colight (Zhang et al., 2022), as well as a pressure-based model trained with PPO (Schulman et al., 2017). The related algorithms are trained and tested in the morning rush hour scenario, from 7:00 to 10:00. The results are presented in Table 2.

Table 2: The results for the traffic signal control scenario with normal traffic conditions.

| Method | Beijing-N | | Shanghai-N | | Paris-N | | New York-N | |
|---|---|---|---|---|---|---|---|---|
| | ATT | TP | ATT | TP | ATT | TP | ATT | TP |
| FixedTime | 4,843.1 | 120,049 | 4,324.1 | 131,020 | 4,245.9 | 60,020 | 4,682.1 | 72,725 |
| MaxPressure | 4,580.4 | 132,055 | **4,045.8** | **144,640** | 3,984.2 | 64,405 | 4,309.9 | 83,927 |
| FRAP | 5,105.2 | 112,321 | 4,671.5 | 121,896 | 4,404.2 | 58,481 | 5,002.5 | 68,196 |
| MPLight | 4,790.1 | 124,991 | 4,674.9 | 122,198 | **3,980.1** | **64,921** | **4,196.4** | **85,331** |
| CoLight | 5,108.9 | 112,672 | 4,640.5 | 124,565 | 4,413.9 | 58,513 | 4,989.2 | 68,184 |
| Efficient-MPLight | 5,101.1 | 113,272 | 4,480.3 | 132,995 | 4,364.9 | 60,428 | 5,019.9 | 67,303 |
| Advanced-MPLight | 5,049.5 | 117,568 | 4,603.6 | 127,911 | 4,281.7 | 61,470 | 5,031.1 | 67,322 |
| Advanced-CoLight | 5,107.2 | 112,778 | 4,661.1 | 123,336 | 4,408.1 | 58,929 | 5,014.7 | 68,292 |
| PPO | **4,452.1** | **136,630** | 4,143.2 | 141,768 | 4,017.6 | 64,277 | 4,254.9 | 84,411 |

**Dynamic Lane Assignment within Junctions Benchmark.** In this scenario, the task is to assign the direction, e.g. left or straight, for the in-going lanes of each junction. We compared the following methods: 1) NoChange, where we keep lane direction unchanged, 2) Random, where we randomly change the direction in every period, 3) Rule, where we estimate the number of vehicles going for each direction and choosing the direction with the maximum number of vehicles, 4) PPO, where we use a PPO-trained model to estimate the number of vehicles. The above algorithms are trained and tested in the morning rush hour scenario, from 7:00 to 10:00. The results are presented in Table 3.

Table 3: The results for the dynamic lane assignment scenario with normal traffic conditions.

| Method | Beijing-N | | Shanghai-N | | Paris-N | | New York-N | |
|---|---|---|---|---|---|---|---|---|
| | ATT | TP | ATT | TP | ATT | TP | ATT | TP |
| NoChange | 4,846.7 | 119,890 | 4,322.2 | 131,145 | 4,245.8 | 60,020 | 4,674.1 | 72,870 |
| Random | 4,839.7 | 120,338 | 4,325.0 | 131,216 | 4,176.7 | 61,810 | 4,636.1 | 74,055 |
| Rule | **4,761.3** | **123,673** | 4,258.2 | 133,346 | **4,155.1** | **62,366** | 4,615.0 | **74,254** |
| PPO | 4,770.0 | 122,929 | **4,256.5** | **133,379** | 4,160.9 | 61,792 | **4,614.5** | 73,907 |

**Tidal Lane Control Benchmark.** In this scenario, the task is to switch the direction of the tidal lane to be forward or backward. We compared the following methods: 1) NoChange, where we disable the tidal lane, 2) Random, where we randomly change the direction in every period, 3) Rule, where we count the number of vehicles going in each direction and choosing the direction with the maximum number of vehicles, 4) PPO, where we use a PPO-trained model to estimate the number of vehicles. The above algorithms are trained and tested in the morning rush hour scenario, from 7:00 to 10:00. The results are presented in Table 4.

Table 4: The results for the tidal lane control scenario with normal traffic conditions.

| Method | Beijing-N | | Shanghai-N | | Paris-N | | New York-N | |
|---|---|---|---|---|---|---|---|---|
| | ATT | TP | ATT | TP | ATT | TP | ATT | TP |
| NoChange | 4,844.6 | 120,105 | 4,334.8 | 130,604 | 4,224.9 | 60,794 | 4,675.1 | 72,834 |
| Random | 4,827.8 | 120,778 | 4,338.4 | 130,283 | 4,216.2 | 60,946 | 4,665.8 | 73,335 |
| Rule | 4,823.4 | 120,901 | 4,313.5 | 131,315 | 4,192.9 | 61,636 | 4,638.0 | 74,284 |
| PPO | **4,820.5** | **120,936** | **4,304.3** | **132,167** | **4,187.4** | **61,738** | **4,628.5** | **74,756** |

**Congestion Pricing Benchmark.** In this scenario, each driver has three candidate routes and the task is to set the price of each road to motivate drivers to choose the route that avoids congested areas. We compared $\Delta$-toll (Sharon et al., 2017) and EBGtoll (Qiu et al., 2019) with two baselines: 1) NoChange, where we do not set the prices, 2) Random, where the drivers randomly choose a route. The above algorithms are trained and tested in the morning rush hour scenario, from 7:00 to 10:00. The results are presented in Table 5.

Table 5: The results for the congestion pricing scenario with normal traffic conditions.

| Method | Beijing-N | | Shanghai-N | | Paris-N | | New York-N | |
|---|---|---|---|---|---|---|---|---|
| | ATT | TP | ATT | TP | ATT | TP | ATT | TP |
| NoChange | 4,840.2 | 120,207 | 4,328.3 | 130,621 | 4,239.2 | 60,100 | 4,681.8 | 72,765 |
| Random | 5,190.4 | 105,640 | 4,422.0 | 131,346 | 4,144.5 | 62,284 | 4,830.9 | 68,976 |
| $\Delta$-toll | **4,667.8** | **131,747** | **4,096.6** | **147,533** | **4,040.7** | **65,024** | **4,549.3** | **78,182** |
| EBGtoll | 5,637.3 | 80,476 | 4,637.6 | 116,624 | 4,240.8 | 60,362 | 5,096.6 | 59,154 |

**Road Planning Benchmark.** In this scenario, the algorithms are asked to select at most 30 roads from 50 candidates for construction to minimize post-construction ATT. We compared 5 methods: 1) NoChange, no of these 50 roads are built, 2) Random, where we select random roads to build, 3) Rule-based, where we select the top-30 vehicle count roads to build, 4) simulated annealing (SA) (Kirkpatrick et al., 1983), 5) bayesian optimization named GeneralBO (Cowen-Rivers et al., 2020). The above algorithms are tested both on morning peak from 6:00 to 12:00 and evening peak from 17:00 to 23:00 and computes the mean of metrics. The result are presented in Table 6.

Table 6: The results for the road planning scenario with normal traffic conditions.

| Method | Beijing-N | | Shanghai-N | | Paris-N | | New York-N | |
|---|---|---|---|---|---|---|---|---|
| | ATT | TP | ATT | TP | ATT | TP | ATT | TP |
| No-Change | 8,439.4 | 161,722 | 6,699.7 | 161,722 | 7,193.9 | **7,632**7 | 7,892.5 | 105,533 |
| Random | 8,304.7 | 163,148 | **6,567.1** | 181,063 | **7,106.2** | 75,839 | 7,967.2 | 102,996 |
| Rule | **8,235.5** | **164,247** | 6,570.7 | **181,504** | 7,177.9 | 76,176 | 7,956.1 | 102,951 |
| SA | 8,332.4 | 163,660 | 6,590.7 | 180,954 | 7,154.2 | 76,164 | 7,871.2 | 105,507 |
| GeneralBO | 8,242.2 | 164,182 | 6,721.8 | 178,790 | 7,161.7 | 75,979 | **7,759.9** | **106,883** |

## 6 CONCLUSION

In this paper, we propose a high-performance large-scale microscopic simulator powered by GPU for transportation system simulation and optimization. We also benchmarked the effect of different optimization algorithms on five transportation system optimization scenarios with different traffic flows in four cities. Interested researchers can use the same pipeline to benchmark most cities around the world with our open source simulator and toolchain. To help interested researchers quickly try out the simulator, we also build a web platform introduced in Appendix H to support no-code trials. We believe that the proposed simulator and platform will contribute to more researchers joining the research work on urban transportation system optimization. We hope that this will not only support more research work on transportation system optimization scenarios, but also promote the development of urban transportation systems towards AI-driven intelligent transportation systems.

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

## A STATEMENT OF OPEN-SOURCE CODES

All the codes with documentation including the simulator, the toolchain, the benchmark codes are open-source and available at Github. However, due to ICLR's double-blind policy, we will not publicize their links in the paper as well as the documentation page. Instead, we anonymize the codes and publish the anonymized links. The anonymized codes may not compile and run properly due to the anonymization process that removes some of the dependent links, and the codes are used ONLY to demonstrate the authenticity of the contributions in the paper in peer review.

- The simulator: `https://anonymous.4open.science/r/moss-AF45`
- The toolchain: `https://anonymous.4open.science/r/mosstool-0242/`
- The benchmark: `https://anonymous.4open.science/r/moss-benchmark-7E7F`

We are constantly developing the simulator and toolchain, so implementation details such as interface names may change with version updates. Replication of benchmark experiments should always be based on the specified version.

## B SYSTEM DESIGN, IMPLEMENTATION, AND COMPARISON

In the section, we present our system execution process to help the reader understand how GPU acceleration is implemented, and detail the vehicle model implementation and related considerations about regarding balancing computational performance with realism.

### B.1 PRELIMINARY

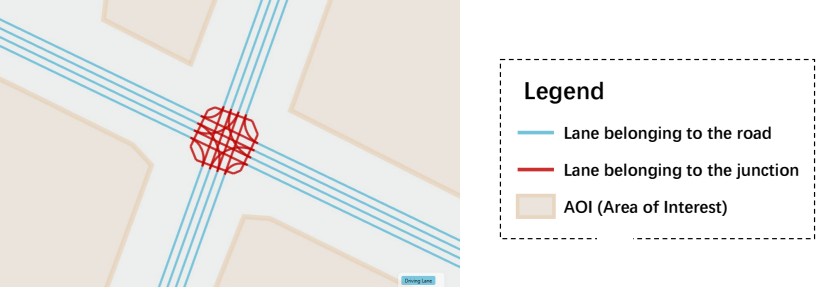

Figure A1: An road network example used for simulation. (best view in color)

To help better understand the system implementation presented in the section, we give the simulation object categories and their definitions as follows.

**Lane.** A lane is the polyline geospatial space in which a vehicle moves and contains center line coordinates. Vehicles always travel along the center line of the lane. There are topological connections between lanes to help the vehicle identify the next lane that can be reached at the end of the current lane travel. Lanes are divided into those belonging to roads and those belonging to junctions as shown in Figure A1. Lanes belonging to a road are parallel to each other without crossing, do not require traffic signals, and allow vehicles to change lanes. In contrast, lanes belonging to a junction may be crossed, requiring a signal and not allowing lane changes.

**Road.** A road is a collection of lanes that are parallel to each other without crossing.

**Junction.** A junction is a collection of lanes that may cross each other and contain a set of traffic signals.

**AOI.** As shown in Figure A1, an area of interest (AOI) is a polygonal geospatial space that has multiple connections to lanes for vehicle entering and exiting. The AOI can be used as a starting and ending location for the vehicle. Alternatively, the start and end locations of the vehicle can be also locations in the lane.

**Person.** Person is a generic term for vehicles, pedestrians, etc. Currently, only vehicles are supported for simulation.

### B.2 EXECUTION PROCESS

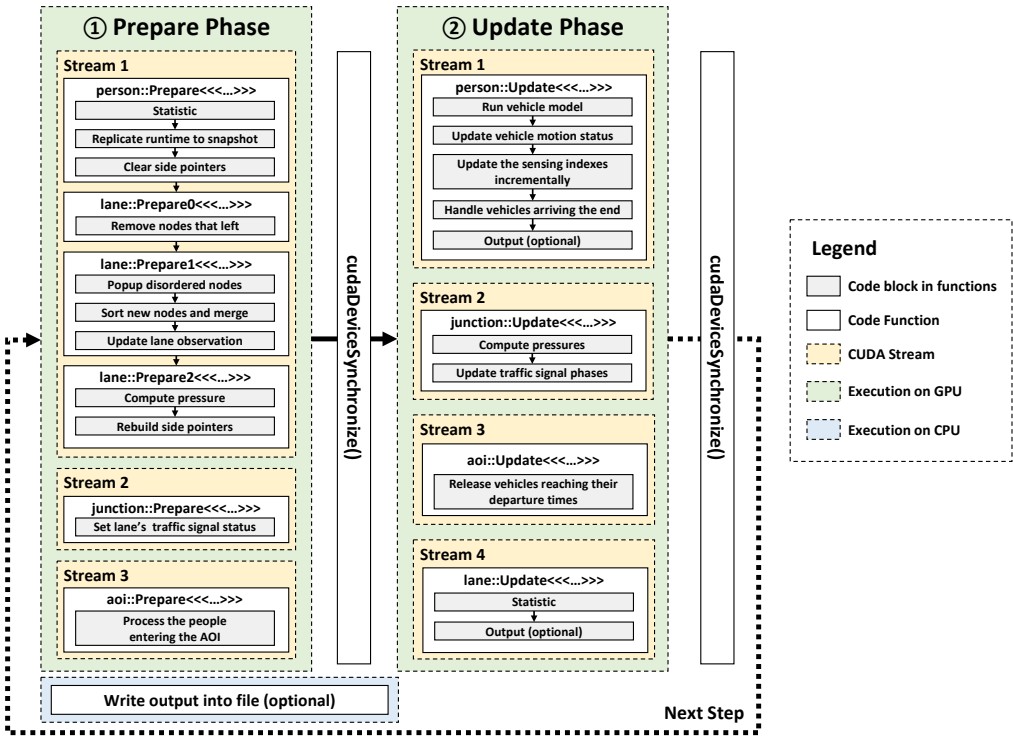

Figure A2: The `Step()` execution process of the proposed simulator. (best view in color)

Same as the baselines, our proposed simulator uses the paradigm of discrete-time simulation for microscopic traffic simulation. The `Step()` function for executing each iteration is displayed in Figure A2. As shown in the figure, all simulation processes are performed on the GPU to avoid the PCI bus data transfer between the CPU and the GPU from becoming a system bottleneck. Based on the two-phase parallel process for read/write separation system design discussed in Section 3.1, the entire iterative process is divided into the *prepare* phase and the *update* phase for sequential execution. The CUDA global synchronization function `cudaDeviceSynchronize()` is executed after each phase to ensure that all tasks launched in that phase are finished. In each stage, in addition to using the

CUDA kernel function (marked as `foo<<<...>>>` in the figure) programming approach directly to perform SIMD calculations, we also use multiple CUDA asynchronous streams to simultaneously execute multiple sets of kernel functions. The above combination maximizes the utilization of GPU hardware power for computational acceleration.

In the *prepare* phase, there are three CUDA asynchronous streams. In Stream 1, we first perform the statistics for vehicle related data through atomic operations and then perform data replication from *runtime* to *snapshot*, which is the key task of the read/write separation strategy in the *prepare* phase. Subsequently, the linked-list based vehicle sensing indexes are updated including clearing side pointers for all nodes, removing nodes that left, reordering, and processing inserted nodes, and finally the side pointers will be rebuilt. To support the other features of the simulation, lane observations are updated so that the AOI knows whether its connection to the lane can put in a vehicle or not, and lane pressures are computed for adaptive traffic signals based on the maximum pressure method. In Stream 2, traffic signal phases at junctions are replicated to the corresponding lanes. In Stream 3, AOIs process people who completed their trip and entered in the previous step.

In the *update* phase, there are four CUDA asynchronous streams. In Stream 1, we perform simulation computations for all people in parallel, including executing vehicle simulation models will be presented in Section B.3, and updating vehicle motion states such as position and velocity based on the actions output from the models. Subsequently, if the lane the vehicle is in changes, the deletion and addition are stored in the buffer of the old lane and the new lane, respectively. If the vehicle reaches the end, it is processed accordingly, e.g., it is added to the AOI addition buffer. All vehicle's motion statuses can be saved for file output if needed. In Stream 2, the junctions collect the pressures of entering and exiting lanes and updates the signal phases using the maximum pressure algorithm. Additionally, signal phase updates can also be made using fixed phases or specified via the Python API. In Stream 3, AOIs put vehicles reaching their departure times into the corresponding lanes if the connections are safe for collision by checking the lane observations. In Stream 4, we perform lane related statistics like average speed and output lane status including traffic signal status.

Besides, we implement the process of writing simulation output into the files on CPU, which becomes a system bottleneck due to both PCI bus bandwidth and disk write speed. However, in reinforcement learning applications, we usually do not need detailed information such as the position and speed of all vehicles at each moment, but rather statistics such as the length of queues at intersections. Also, the data is stored in memory instead of files. Therefore the performance experiments for our proposed simulator and baselines do not include writing simulation results to a file.

## B.3 VEHICLE MODEL IMPLEMENTATION COMPARED TO THE BASELINE SYSTEMS

The implementation of the execution process described above focuses on illustrating how to simulate efficiently, and this subsection focuses on the vehicle simulation model used in our proposed simulator and the performance versus realism trade-offs considered. We will also discuss comparisons and differences among our vehicle simulation models and baseline systems.

In our simulator, the vehicle behavior is controlled by the car-following model, the lane-changing model, and the traffic signal strategy.

For the car-following model, we use the IDM model (Treiber et al., 2000). The formula is as follows:

$$a(t) = a_{max} \left[ 1 - \left( \frac{v(t)}{v_0} \right)^{\delta} - \left( \frac{s^*(t)}{s(t)} \right)^2 \right] \tag{1}$$

$$s^*(t) = s_0 + max \left( 0, v(t) \cdot T + \frac{v(t) \cdot \Delta v(t)}{2\sqrt{a_{max} \cdot a_{comf}}} \right), \tag{2}$$

where $a_{max}$ is the maximum acceleration of the vehicle, $a_{comf}$ is the comfortable deceleration, $T$ is the headway, $s_0$ is the minimum distance from the front vehicle. The four parameters are set in the vehicle's profile. $v_0$ is the expected velocity of the vehicle, that is, the minimum value of road velocity limit and maximum vehicle velocity. $v(t)$, $s(t)$, and $a(t)$ are the velocity, the distance from the front vehicle, the acceleration at the moment $t$, respectively. $\Delta v(t)$ is the difference in velocity between the vehicle and the front one. $\delta$ is a hyper-parameter of the model, which is set $4.0$. In the implementation, the velocity and distance of the front vehicle is obtained by the vehicle sensing

indexes with only one pointer operation. If there is no front vehicle in the lane, the first vehicle in the next lane will be used as a substitution.

For the lane-changing model, we use a randomized MOBIL model. The MOBIL model (Kesting et al., 2007) first calculate a utility by the following formula:

$$u = (\tilde{a}_{ego} - a_{ego}) + p \cdot ((\tilde{a}_{new} - a_{new}) + (\tilde{a}_{old} - a_{old})),$$

where $p$ is the politeness factor which is set 0.1. $\tilde{a}_{ego}$ and $a_{ego}$ denote the original acceleration of the vehicle and its new acceleration if the vehicle changes its lane, respectively. $\tilde{a}_{new}$ and $a_{new}$ denote the original acceleration of the new follower and its new acceleration if the ego vehicle changes its lane, respectively. $\tilde{a}_{old}$ and $a_{old}$ denote the original acceleration of the old follower and its new acceleration if the ego vehicle changes its lane, respectively. Following Feng et al. (2021), we introduce randomization to avoid identical lane-changing behavior of all vehicles and to break perfect symmetry. The total utility is calculated as follows:

$$u_T = max(0, u_L) + max(0, u_R),$$

where $u_L$ and $u_R$ denote the utilities of changing lanes to the left and right lanes, respectively. Then a total lane change probability is computed as follows:

$$p_{LC} = \begin{cases} 0.9, & u_T \geq 1 \\ (0.9 - 2 \times 10^{-8})u_T, & 0 < u_T < 1 \\ 2 \times 10^{-8}, & u_T = 0 \end{cases}$$

The vehicle firstly determines whether to change lanes based on the probability $p_{LC}$, and secondly determines the direction of lane change based on $u_L$ and $u_R$. The all acceleration computation use the car-following model and the inputs are also obtained by the side pointer mechanism in the vehicle sensing indexes. Vehicles are prohibited from changing lanes away from the group of lanes that allow access to the next road.

For traffic signals, the vehicle always checks the status of the signal for the next lane and decelerates to a stop based on the IDM model (assuming a stationary vehicle at the end of the current lane) in a red or yellow light situation.

Table A1: Comparison of key components in vehicle simulation modeling.

| Vehicle Behavior | SUMO | CityFlow | CBLab | Ours |
|---|---|---|---|---|
| Car-following | Selectable | Krauß | Krauß | IDM |
| Lane-changing | | Earliest | Private | Randomized MOBIL |
| Traffic Signal | ✓ | ✓ | ✓ | ✓ |
| Overlap and priority in junction | ✓ | ✓ | - | - |

We compare the key components in vehicle simulation modeling among the baseline systems and our proposed simulator in the above Table A1. *Car-following* and *Lane-changing* are discussed at Section 2.1. *Traffic signal* indicates whether the vehicle is capable of adjusting its speed to the signal status before the junction, which is plain. *Overlap and priority in junction* means whether vehicles within the junction are considered for access prioritization to avoid potential intersecting vehicles. Both SUMO and CityFlow model overlap and priority in junction. However, based on the findings reported in the CBLab article (in Section 2.5.1) with our test results, modeling of overlap and priority in junction can easily lead to vehicle deadlocks (i.e., 3 or more vehicles waiting for each other) under large-scale simulations. Therefore, we adopt the same treatment as CBLab, dropping the modeling of overlap and priority in junction to ensure the practicability of the simulation in large-scale scenarios. In this setup, the realism of the junction simulation relies on conflict-free signal phase settings. This is a limitation of our proposed simulator.

## C DATASETS AND SETTINGS FOR SIMULATOR PERFORMANCE COMPARISON

The statistics of the datasets used for the simulator performance comparison experiments are shown in Table A2. For hardware settings, all simulations are conducted in the same hardware environment

Table A2: The statistics of the datasets used for simulator performance comparison.

| Roadnet | #Road | #Junction | #Vehicle |
|---------|-------|-----------|----------|
| Roadnet-S | 214 | 67 | {19; 194; 1,944} |
| Roadnet-M | 10,502 | 2,929 | {1,145; 11,456; 114,561} |
| Roadnet-L | 93,564 | 26,479 | {24,649; 246,495; 2,464,950} |

with an Intel(R) Xeon(R) Platinum 8462Y CPU (64 threads) and an NVIDIA GeForce RTX 4090 GPU.

## D    VISUALIZATION OF SIMULATOR REALISM COMPARISON

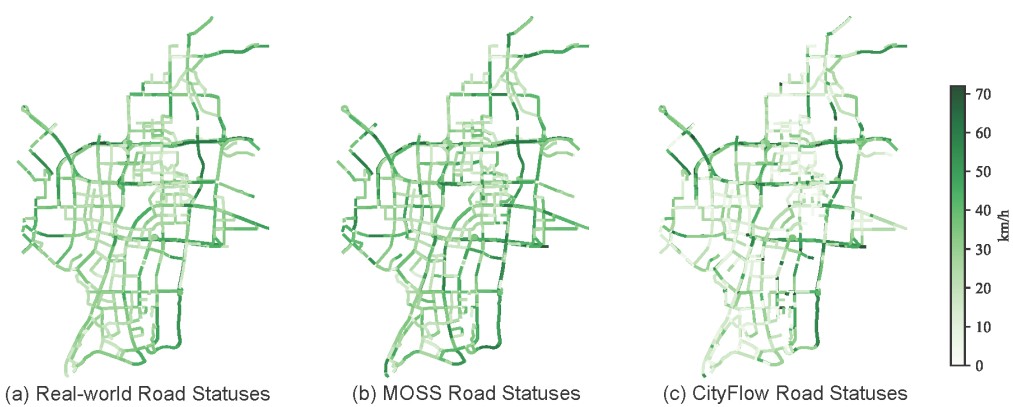

(a) Real-world Road Statuses    (b) MOSS Road Statuses    (c) CityFlow Road Statuses

Figure A3: Visualization and comparison of road traffic status from (a) the Shenzhen dataset, (b) our proposed simulator, and (c) CityFlow. (best view in color)

## E    DATASETS FOR TRANSPORTATION SYSTEM OPTIMIZATION BENCHMARKING

In the section, we describe in detail the process of constructing the datasets used for transportation system optimizaiton benchmarking. We chose 4 representative big cities around the world, including Beijing, Shanghai, Paris and New York, as our targets. We use OpenStreetMap (OSM) [4] as the data source for map construction and a diffusion model based on publicly available data represented by satellite imagery as input to generate realistic travel origin-destination (OD) matrices as travel demands.

**Map Building.** First, based on the toolchain, we selected the bounding boxes as shown in Table A3 for each city for to build the map.

Table A3: Geometry bounding boxes of four cities.

| Bounding Box | maximum latitude | minimum latitude | maximum longitude | minimum longitude |
|--------------|------------------|------------------|-------------------|-------------------|
| Beijing | 40.131 | 39.771 | 116.626 | 116.158 |
| Shanghai | 31.389 | 31.100 | 121.676 | 121.313 |
| Paris | 48.949 | 48.745 | 2.514 | 2.131 |
| New York | 40.941 | 40.567 | -73.697 | -74.058 |

---

[4]https://openstreetmap.org/

Specifically, we first convert OSM data within the bounding boxes to GeoJSON format, and secondly build maps from GeoJSON format data. The statistics of the maps of the four cities are shown in Table A4.

Table A4: Statistics of the four maps.

| Statistics | # of roads | # of junctions |
|---|---|---|
| Beijing | 25,945 | 11,953 |
| Shanghai | 14,837 | 6,270 |
| Paris | 14,411 | 6,588 |
| New York | 19,046 | 8,339 |

**Realistic OD Matrix Generation.** In order to generate realistic travel demands of the four cities, we perform OD matrix generation based on a diffusion model provided in the toolchain that has been pre-trained in several regions around the world.

**Obtaining Travel Demand of Different Congestion Levels.** In order not to introduce too many variables, we assume that driving is used for all trips. In addition, to better represent commuting traffic, we assume that the departure times of all vehicles are limited to the morning and evening peaks. And we scale the total traffic volume to get the travel demand under different congestion levels. For the morning peak, we adjust vehicles' arrival time to create a morning peak flows by using a uniform distribution between 8 o'clock and 9 o'clock. We then subtracted the estimated travel time from the arrival time to get the departure time. The estimated travel time is calculated by dividing the route length by the vehicle speed, which is set as $60km/h$ for our experiment.

Similarly, we create an evening peak group by exchanging the origin and destination of individuals from the morning peak flows. Their departure times are set to be uniformly distributed between 17 o'clock and 18 o'clock.

After completing the above steps, we use A* algorithm to calculate the route based on the shortest time and assign it to each vehicles and remove those who are unable to reach their destinations. For each city, we scale the number of vehicles and observe the arrival rate of all vehicles, which refers to the rate of vehicles that successfully reached their destinations, to construct datasets with different congestion levels.

The arrival rate of the dataset is determined as the minimum rate between the morning and evening peak periods. Specifically, an arrival rate of $80\%$ is considered congested, $90\%$ is considered normal, and $95\%$ is considered smooth. Based on the above rates, we construct the travel demand datasets under different congestion levels in the four cities, and the relevant statistics are shown in Table A5.

Table A5: #trips of the datasets with different congestion level.

| Congestion Level | Smooth | Normal | Congested |
|---|---|---|---|
| Beijing | 350,838 | 439,280 | 571,412 |
| Shanghai | 348,880 | 436,888 | 612,160 |
| Paris | 154,276 | 202,664 | 251,236 |
| New York | 218,712 | 262,706 | 306,078 |

# F THE SETTINGS OF TRANSPORTATION SYSTEM OPTIMIZATION SCENARIOS

All experiments are conducted in the same hardware environment with an Intel(R) Xeon(R) Platinum 8462Y CPU (64 threads) and an NVIDIA GeForce RTX 4090 GPU. The training time varies across different scenarios. The optimal hyper-parameters are grid-searched and hard-coded into the released code. Please refer to the release files for detailed hyper-parameter settings.

### F.1 Traffic Signal Control.

**Scenario.** There are multiple junctions in the road network with traffic signals to be controlled. Each junction has a list of available traffic signal phases predefined according to the geometry of the junction, like the number and direction of the incoming and outgoing lanes. Every $T = 30$ seconds, the agent has to choose one phase from the list to be applied in the next period.

**Observation.** The observation includes the geometry of the junction and the number of (all/waiting) vehicles on each lane.

**Action.** Choose one phase from the given list.

**Reward.** Opposite of the average number of waiting vehicles on the incoming lanes.

**Training.** The learning-based methods are all trained for 4 hours.

### F.2 Dynamic Lane Assignment within Junctions.

**Scenario.** There are multiple roads in the road network with dynamic lanes at the end where the roads connect to junctions. Each road has exactly one dynamic lane whose direction can be either LEFT or STRAIGHT. Every $T = 30$ seconds, the agent has to assign the direction of the dynamic lane.

**Observation.** The observation includes the geometry of the junction and the number of (all/waiting) vehicles on each lane.

**Action.** Choose one of the two directions.

**Reward.** Opposite of the average number of waiting vehicles on the lanes of the road.

**Training.** The learning-based methods are all trained for 3 hours.

### F.3 Tidal Lane Control.

**Scenario.** There are multiple road pairs in the road network with tidal lanes. Each road pair has exactly one tidal lane in the center whose direction can be either FORWARD or BACKWARD. Every $T = 180$ seconds, the agent has to choose the direction of the tidal lane.

**Observation.** The observation includes the geometry of the road and the number of (all/waiting) vehicles on each lane.

**Action.** Choose one of the two directions.

**Reward.** Opposite of the average number of waiting vehicles on the lanes of the road.

**Training.** The learning-based methods are all trained for 3 hours.

### F.4 Congestion Pricing.

**Scenario.** All the roads in the road network can be set with a congestion price for vehicles traveling through it. Every $T = 20$ seconds, the agent can change the prices according to the traffic condition.

**Observation.** The observation includes the geometry of the road network and the number of (all/waiting) vehicles on each lane.

**Action.** Set the prices for each road.

**Reward.** The number of finished vehicles in the past period.

**Training.** The learning-based methods are all trained for 3 hours.

### F.5 Road Planning.

**Scenario.** In the road network, there are multiple newly constructed roads during the past five years. Each of these roads has two statuses, either KEEP or REMOVE. The algorithms observe the ATT and are asked to minimize the ATT by setting the road statues as KEEP or REMOVE.

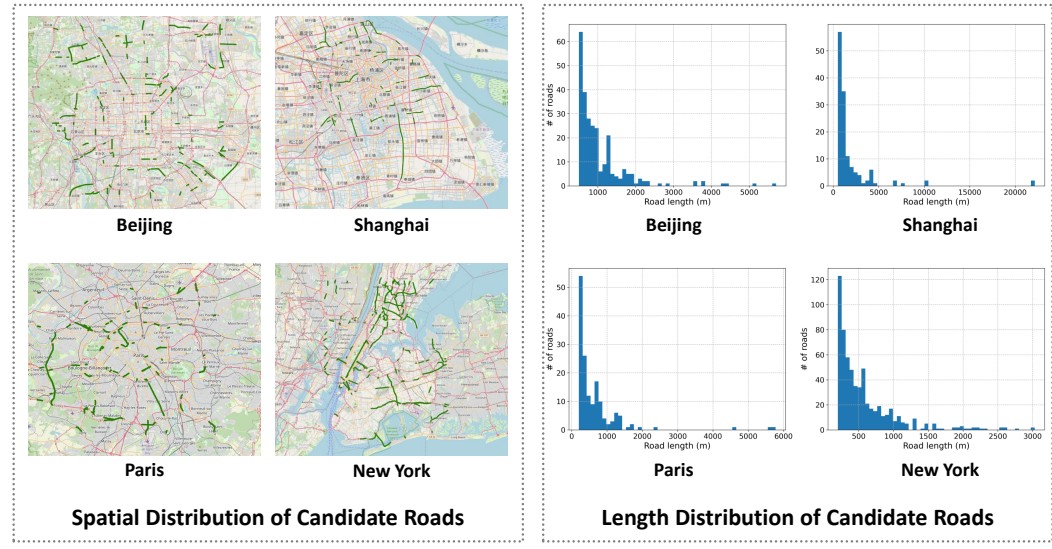

Figure A4: The spatial and length distribution of all candidate roads.

**Candidate Roads Identification.** For each city, we extract driving roads from OSM of 2019. We match every road in our map to the road network of 2019, there are three aspects to evaluate the matching, the distance between two roads, the distance between the middle point of road in our map and difference highway level between two roads. Any road that cannot be matched with any roads in 2019 is identified as a newly constructed road, and regarded as a candidate road. The spatial and length statistics of candidate roads are shown in Figure A4. We select the 50 roads with the highest number of vehicles from each candidate set as the optimization set for the algorithm.

Table A6: # of candidate roads.

| Basic Statistics | # of roads |
|---|---|
| Beijing | 263 |
| Shanghai | 136 |
| Paris | 156 |
| New York | 612 |

## G COMPLETE BENCHMARK RESULTS

### G.1 TRAFFIC SIGNAL CONTROL.

Table A7: The results for the traffic signal control scenario with smooth traffic conditions.

| Method | Beijing-S | | Shanghai-S | | Paris-S | | New York-S | |
|---|---|---|---|---|---|---|---|---|
| | ATT | TP | ATT | TP | ATT | TP | ATT | TP |
| FixedTime | 4,426.5 | 109,588 | 3,878.6 | 120,449 | 3,749.7 | 52,459 | 4,369.4 | 67,046 |
| MaxPressure | 4,099.1 | 122,737 | 3,523.4 | 132,028 | **3,477.4** | 55,123 | 3,951.8 | 77,018 |
| FRAP | 4,741.0 | 102,556 | 4,261.8 | 113,613 | 3,920.7 | 51,301 | 4,725.9 | 63,572 |
| MPLight | 4,087.0 | 121,620 | **3,474.9** | **133,044** | 3,486.6 | **55,946** | **3,840.8** | **78,125** |
| CoLight | 4,757.3 | 102,433 | 4,221.9 | 115,244 | 3,922.7 | 51,351 | 4,714.3 | 63,553 |
| Efficient-MPLight | 4,529.5 | 110,199 | 4,013.2 | 121,301 | 3,953.4 | 51,236 | 4,390.9 | 69,279 |
| Advanced-MPLight | 4,750.3 | 102,649 | 3,997.8 | 120,529 | 3,786.5 | 53,982 | 4,378.7 | 70,352 |
| Advanced-CoLight | 4,740.8 | 103,172 | 4,245.1 | 114,927 | 3,945.8 | 51,164 | 4,748.6 | 63,463 |
| PPO | **4,005.7** | **124,001** | 3,636.3 | 130,485 | 3,485.3 | 55,915 | 3,914.8 | 77,007 |

Table A8: The results for the traffic signal control scenario with normal traffic conditions.

| Method | Beijing-N | | Shanghai-N | | Paris-N | | New York-N | |
|---|---|---|---|---|---|---|---|---|
| | ATT | TP | ATT | TP | ATT | TP | ATT | TP |
| FixedTime | 4,843.1 | 120,049 | 4,324.1 | 131,020 | 4,245.9 | 60,020 | 4,682.1 | 72,725 |
| MaxPressure | 4,580.4 | 132,055 | **4,045.8** | **144,640** | 3,984.2 | 64,405 | 4,309.9 | 83,927 |
| FRAP | 5,105.2 | 112,321 | 4,671.5 | 121,896 | 4,404.2 | 58,481 | 5,002.5 | 68,196 |
| MPLight | 4,790.1 | 124,991 | 4,674.9 | 122,198 | **3,980.1** | **64,921** | 4,196.4 | 85,331 |
| CoLight | 5,108.9 | 112,672 | 4,640.5 | 124,565 | 4,413.9 | 58,513 | 4,989.2 | 68,184 |
| Efficient-MPLight | 5,101.1 | 113,272 | 4,480.3 | 132,995 | 4,364.9 | 60,428 | 5,019.9 | 67,303 |
| Advanced-MPLight | 5,049.5 | 117,568 | 4,603.6 | 127,911 | 4,281.7 | 61,470 | 5,031.1 | 67,322 |
| Advanced-CoLight | 5,107.2 | 112,778 | 4,661.1 | 123,336 | 4,408.1 | 58,929 | 5,014.7 | 68,292 |
| PPO | **4,452.1** | **136,630** | 4,143.2 | 141,768 | 4,017.6 | 64,277 | 4,254.9 | 84,411 |

Table A9: The results for the traffic signal control scenario with congested traffic conditions.

| Method | Beijing-C | | Shanghai-C | | Paris-C | | New York-C | |
|---|---|---|---|---|---|---|---|---|
| | ATT | TP | ATT | TP | ATT | TP | ATT | TP |
| FixedTime | 5,274.9 | 130,419 | 4,928.6 | 145,308 | 4,587.8 | 66,608 | 4,923.7 | 76,165 |
| MaxPressure | 5,106.5 | 142,654 | 4,790.7 | 158,804 | **4,371.4** | 71,874 | 4,596.3 | 87,897 |
| FRAP | 5,490.9 | 122,061 | 5,205.6 | 133,039 | 4,743.1 | 64,711 | 5,219.1 | 71,448 |
| MPLight | 5,496.3 | 121,452 | **4,740.9** | **160,551** | 4,378.1 | **72,643** | **4,468.6** | **90,632** |
| CoLight | 5,484.5 | 122,508 | 5,185.2 | 135,131 | 4,750.7 | 64,463 | 5,214.9 | 72,091 |
| Efficient-MPLight | 5,375.6 | 130,151 | 5,104.4 | 142,562 | 4,718.1 | 65,795 | 5,225.2 | 71,556 |
| Advanced-MPLight | 5,358.1 | 129,512 | 5,129.9 | 138,727 | 4,650.8 | 67,593 | 5,227.9 | 71,713 |
| Advanced-CoLight | 5,481.2 | 122,564 | 5,207.1 | 133,709 | 4,756.0 | 64,517 | 5,222.8 | 71,697 |
| PPO | **4,975.5** | **149,589** | 4,814.6 | 155,279 | 4,394.6 | 71,491 | 4,536.0 | 90,111 |

## G.2 DYNAMIC LANE ASSIGNMENT WITHIN JUNCTIONS.

Table A10: The results for the dynamic lane assignment scenario with smooth traffic conditions.

| Method | Beijing-S | | Shanghai-S | | Paris-S | | New York-S | |
|---|---|---|---|---|---|---|---|---|
| | ATT | TP | ATT | TP | ATT | TP | ATT | TP |
| NoChange | 4,426.4 | 109,588 | 3,878.6 | 120,449 | 3,749.6 | 52,459 | 4,369.5 | 67,046 |
| Random | 4,435.4 | 109,450 | 3,875.3 | 120,896 | 3,681.6 | 53,474 | 4,344.2 | 67,271 |
| Rule | **4,340.6** | **112,462** | 3,807.6 | 122,688 | 3,675.5 | 53,244 | 4,310.5 | **67,737** |
| PPO | 4,345.0 | 111,391 | **3,804.2** | **122,692** | **3,663.4** | **53,629** | **4,309.7** | 67,626 |

Table A11: The results for the dynamic lane assignment scenario with normal traffic conditions.

| Method | Beijing-N | | Shanghai-N | | Paris-N | | New York-N | |
|---|---|---|---|---|---|---|---|---|
| | ATT | TP | ATT | TP | ATT | TP | ATT | TP |
| NoChange | 4,846.7 | 119,890 | 4,322.2 | 131,145 | 4,245.8 | 60,020 | 4,674.1 | 72,870 |
| Random | 4,839.7 | 120,338 | 4,325.0 | 131,216 | 4,176.7 | 61,810 | 4,636.1 | 74,055 |
| Rule | **4,761.3** | **123,673** | 4,258.2 | 133,346 | **4,155.1** | **62,366** | 4,615.0 | **74,254** |
| PPO | 4,770.0 | 122,929 | **4,256.5** | **133,379** | 4,160.9 | 61,792 | **4,614.5** | 73,907 |

Table A12: The results for the dynamic lane assignment scenario with congested traffic conditions.

| Method | Beijing-C | | Shanghai-C | | Paris-C | | New York-C | |
|---|---|---|---|---|---|---|---|---|
| | ATT | TP | ATT | TP | ATT | TP | ATT | TP |
| NoChange | 5,275.0 | 130,419 | 4,928.5 | 145,308 | 4,587.8 | 66,608 | 4,923.8 | 76,165 |
| Random | 5,294.3 | 129,976 | 4,932.6 | 144,774 | 4,539.3 | 68,410 | 4,907.5 | 77,460 |
| Rule | **5,221.1** | **134,661** | 4,875.0 | **148,393** | **4,521.8** | **68,795** | 4,863.8 | 78,469 |
| PPO | 5,240.1 | 131,854 | **4,874.5** | 148,228 | 4,522.4 | 68,413 | **4,854.7** | **78,857** |

## G.3 TIDAL LANE CONTROL.

Table A13: The results for the tidal lane control scenario with smooth traffic conditions.

| Method | Beijing-S | | Shanghai-S | | Paris-S | | New York-S | |
|---|---|---|---|---|---|---|---|---|
| | ATT | TP | ATT | TP | ATT | TP | ATT | TP |
| NoChange | 4,422.7 | 109,912 | 3,884.8 | 120,156 | 3,723.2 | 52,992 | 4,388.5 | 66,269 |
| Random | 4,418.9 | 110,150 | 3,887.8 | 119,940 | 3,711.3 | 53,313 | 4,360.2 | 67,295 |
| Rule | 4,416.8 | 109,997 | 3,870.3 | 120,366 | 3,687.6 | 53,626 | 4,341.3 | 68,113 |
| PPO | **4,411.0** | **110,161** | **3,857.9** | **121,344** | **3,674.5** | **53,922** | **4,325.1** | **68,775** |

Table A14: The results for the tidal lane control scenario with normal traffic conditions.

| Method | Beijing-N | | Shanghai-N | | Paris-N | | New York-N | |
|---|---|---|---|---|---|---|---|---|
| | ATT | TP | ATT | TP | ATT | TP | ATT | TP |
| NoChange | 4,844.6 | 120,105 | 4,334.8 | 130,604 | 4,224.9 | 60,794 | 4,675.1 | 72,834 |
| Random | 4,827.8 | 120,778 | 4,338.4 | 130,283 | 4,216.2 | 60,946 | 4,665.8 | 73,335 |
| Rule | 4,823.4 | 120,901 | 4,313.5 | 131,315 | 4,192.9 | 61,636 | 4,638.0 | 74,284 |
| PPO | **4,820.5** | **120,936** | **4,304.3** | **132,167** | **4,187.4** | **61,738** | **4,628.5** | **74,756** |

Table A15: The results for the tidal lane control scenario with congested traffic conditions.

| Method | Beijing-C | | Shanghai-C | | Paris-C | | New York-C | |
|---|---|---|---|---|---|---|---|---|
| | ATT | TP | ATT | TP | ATT | TP | ATT | TP |
| NoChange | 5,278.4 | 130,133 | 4,937.9 | 144,546 | 4,575.6 | 67,201 | 4,923.7 | 76,253 |
| Random | 5,274.2 | 130,456 | 4,937.8 | 143,921 | 4,568.5 | 67,661 | 4,903.8 | 77,236 |
| Rule | 5,260.0 | 131,304 | **4,908.7** | 146,094 | 4,555.5 | 68,025 | 4,877.9 | 79,096 |
| PPO | **5,258.7** | **131,709** | 4,909.5 | **146,145** | **4,544.7** | **68,274** | **4,860.9** | **79,731** |

## G.4 CONGESTION PRICING.

Table A16: The results for the congestion pricing scenario with smooth traffic conditions.

| Method | Beijing-S | | Shanghai-S | | Paris-S | | New York-S | |
|---|---|---|---|---|---|---|---|---|
| | ATT | TP | ATT | TP | ATT | TP | ATT | TP |
| NoChange | 4,433.4 | 109,604 | 3,875.9 | 120,444 | 3,743.2 | 52,611 | 4,368.1 | 66,984 |
| Random | 4,865.6 | 96,231 | 3,969.0 | 121,137 | 3,705.2 | 53,904 | 4,571.7 | 63,670 |
| $\Delta$-toll | **4,267.1** | **118,077** | **3,630.4** | **133,056** | **3,611.0** | **54,762** | **4,246.6** | **72,188** |
| EBGtoll | 5,348.1 | 75,078 | 4,230.1 | 107,683 | 3,759.7 | 52,424 | 4,837.9 | 55,155 |

Table A17: The results for the congestion pricing scenario with normal traffic conditions.

| Method | Beijing-N | | Shanghai-N | | Paris-N | | New York-N | |
|---|---|---|---|---|---|---|---|---|
| | ATT | TP | ATT | TP | ATT | TP | ATT | TP |
| NoChange | 4,840.2 | 120,207 | 4,328.3 | 130,621 | 4,239.2 | 60,100 | 4,681.8 | 72,765 |
| Random | 5,190.4 | 105,640 | 4,422.0 | 131,346 | 4,144.5 | 62,284 | 4,830.9 | 68,976 |
| $\Delta$-toll | **4,667.8** | **131,747** | **4,096.6** | **147,533** | **4,040.7** | **65,024** | **4,549.3** | **78,182** |
| EBGtoll | 5,637.3 | 80,476 | 4,637.6 | 116,624 | 4,240.8 | 60,362 | 5,096.6 | 59,154 |

Table A18: The results for the congestion pricing scenario with congested traffic conditions.

| Method | Beijing-C | | Shanghai-C | | Paris-C | | New York-C | |
|---|---|---|---|---|---|---|---|---|
| | ATT | TP | ATT | TP | ATT | TP | ATT | TP |
| NoChange | 5,281.9 | 129,904 | 4,927.9 | 145,154 | 4,588.8 | 66,603 | 4,928.0 | 76,140 |
| Random | 5,558.6 | 115,184 | 5,027.3 | 143,400 | 4,488.2 | 69,075 | 5,066.6 | 72,916 |
| $\Delta$-toll | **5,141.4** | **143,518** | **4,767.7** | **162,829** | **4,399.9** | **72,309** | **4,801.0** | **82,602** |
| EBGtoll | 5,912.5 | 86,945 | 5,182.9 | 128,301 | 4,571.6 | 66,971 | 5,275.4 | 63,052 |

## G.5 ROAD PLANNING.

Table A19: The results for the road planning scenario with smooth traffic conditions.

| Method | Beijing-S | | Shanghai-S | | Paris-S | | New York-S | |
|---|---|---|---|---|---|---|---|---|
| | ATT | TP | ATT | TP | ATT | TP | ATT | TP |
| No-Change | 7,411.5 | 137,793 | 5,432.1 | 151,689 | 5,997.5 | 62,016 | 6,954.1 | 92,756 |
| Random | 7,216.4 | 139,070 | 5,325.6 | 151,828 | 5,899.9 | 62,070 | 7,176.5 | 88,947 |
| Rule | 7,240.7 | 138,795 | 5,352.5 | **152,124** | 5,907.3 | 62,155 | 6,942.7 | 92,463 |
| SA | 7,272.0 | 139,326 | 5,358.9 | 151,983 | 5,772.8 | **62,990** | 6,921.1 | 92,754 |
| GeneralBO | **7,102.8** | **139,983** | **5,297.7** | 152,046 | **5,755.4** | 62,478 | **6,808.2** | **93,858** |

Table A20: The results for the road planning scenario with normal traffic conditions.

| Method | Beijing-N | | Shanghai-N | | Paris-N | | New York-N | |
|---|---|---|---|---|---|---|---|---|
| | ATT | TP | ATT | TP | ATT | TP | ATT | TP |
| No-Change | 8,439.4 | 161,722 | 6,699.7 | 161,722 | 7,193.9 | **7,632**7 | 7,892.5 | 105,533 |
| Random | 8,304.7 | 163,148 | **6,567.1** | 181,063 | **7,106.2** | 75,839 | 7,967.2 | 102,996 |
| Rule | **8,235.5** | **164,247** | 6,570.7 | **181,504** | 7,177.9 | 76,176 | 7,956.1 | 102,951 |
| SA | 8,332.4 | 163,660 | 6,590.7 | 180,954 | 7,154.2 | 76,164 | 7,871.2 | 105,507 |
| GeneralBO | 8,242.2 | 164,182 | 6,721.8 | 178,790 | 7,161.7 | 75,979 | **7,759.9** | **106,883** |

Table A21: The results for the road planning scenario with congested traffic conditions.

| Method | Beijing-C | | Shanghai-C | | Paris-C | | New York-C | |
|---|---|---|---|---|---|---|---|---|
| | ATT | TP | ATT | TP | ATT | TP | ATT | TP |
| No-Change | 9,684.4 | 191,786 | 8,793.1 | 216,625 | 8,310.1 | 87,786 | 8,624.6 | 117,606 |
| Random | 9,496.8 | 195,463 | 8,652.0 | 219,199 | 8,230.8 | 87,704 | 8,663.1 | 115,278 |
| Rule | 9,489.2 | 195,427 | 8,656.6 | 218,214 | 8,250.9 | 87,872 | 8,682.7 | 114,650 |
| SA | 9,487.0 | 195,918 | 8,600.0 | 219,711 | 8,188.4 | 88,650 | 8,589.7 | 118,036 |
| GeneralBO | **9,369.1** | **198,446** | **8,508.8** | **221,049** | **8,161.1** | **88,803** | **8,542.2** | **118,579** |

# H    WEB PLATFORM GUIDANCE

Since the simulation and optimization of urban transportation systems needs to include map construction, travel demand generation, simulation, and optimization, its requires interested researchers to invest their time in learning and writing the relevant code. In order to help researchers quickly try out the simulator and simulate and optimize urban transportation systems in any region of the world, we build a web platform to support no-code trials. This platform is open for registration and contains mainly the wizard program with documentation for the simulator as shown in Figure A5.

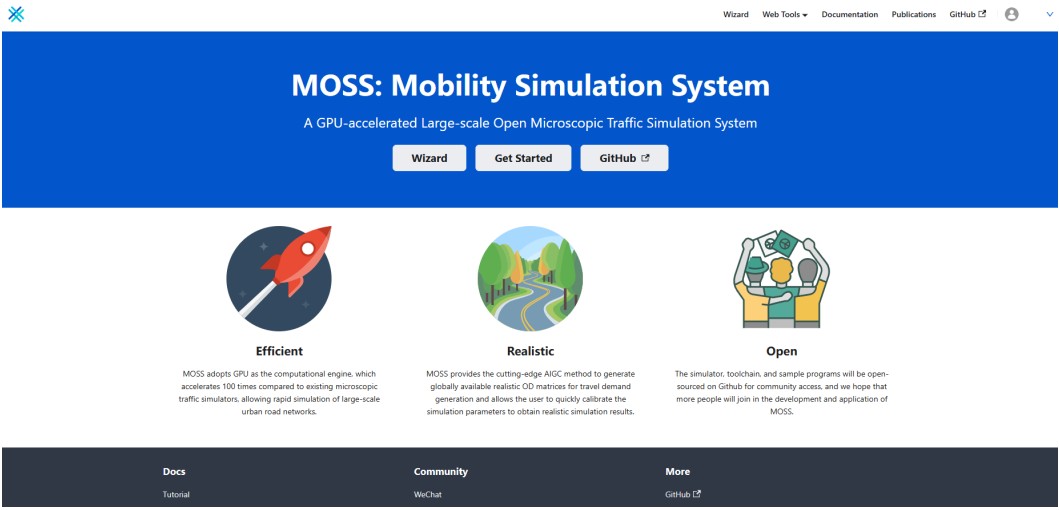

Figure A5: The home page of the web platform. (best view in color)

Upon entering the **Wizard** page, the platform provides a complete process of building maps, generating trips, simulation, and optimization.

As shown in Figure A6, you can select any rectangular area on the world map or input the bounding box to submit a map building task. The platform is also pre-built with some of the world's major cities to select from. You can download the map binary file as the simulation input or continue to use the online platform to generate trip by clicking the **Select & Continue** button.

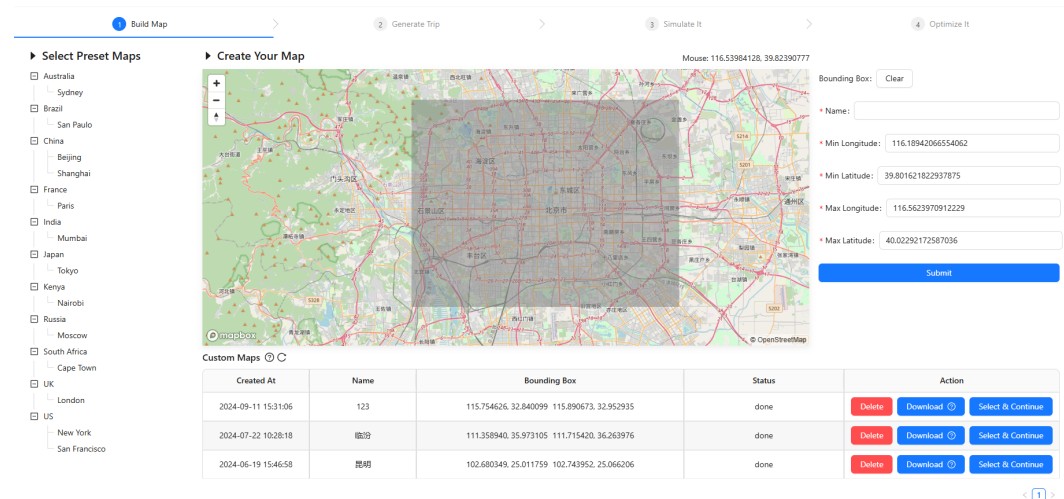

Figure A6: The map building page of the web platform. (best view in color)

After building or selecting the map, the second step is to generate the travel demands named as trips in the platform as shown in Figure A7. Following the travel demand construction methodology used in the article as mentioned in Section E, we provide a diffusion model based OD matrix generation method named as AIGC. You can customize the area grid density and total number of people used for OD matrix generation, which will affect the time required for execution. You can also customize the departure time distribution through the web GUI. In addition to OD generation, the platform also automatically performs route selection based on minimum elapsed time to generate the correct inputs that can be used directly for simulation. The results can also download for local execution. Once you have finished generating trips, you can click **Select & Continue** to access the simulation page.

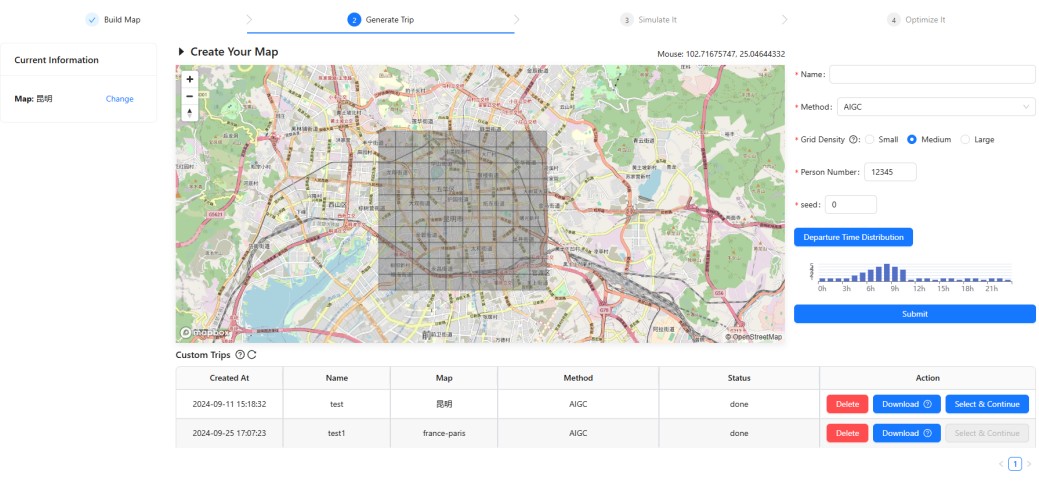

Figure A7: The trip generation page of the web platform. (best view in color)

As shown in Figure A8, you can start the simulation by simply setting the time range to be simulated. Once the simulation is complete, you can click the **Show Result** button to view the simulation results. The visualization of the simulation results is based on the WebGL online map rendering framework, in which you can view the movement of vehicles, changes in junction traffic signals, etc., as well as observe the automatically constructed road network and its connectivity.

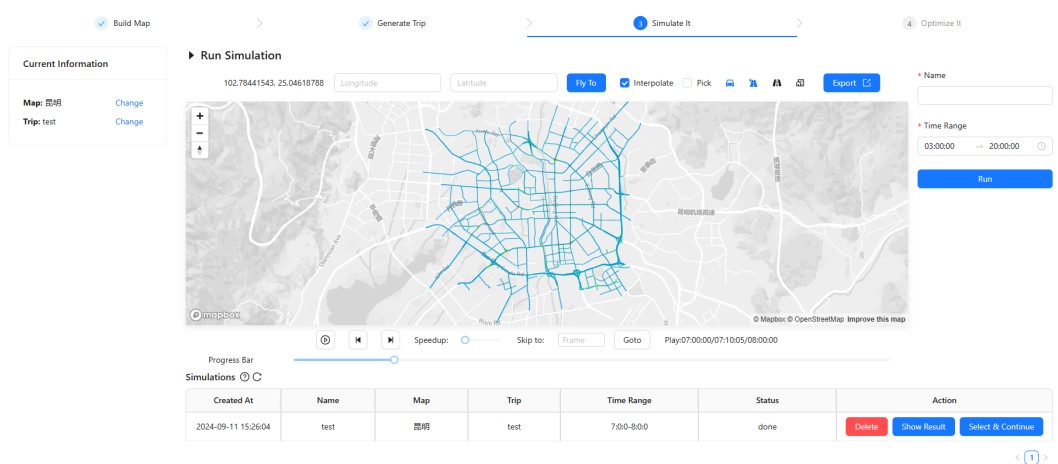

Figure A8: The simulation page of the web platform. (best view in color)

As shown in Figure A9, the online platform also supports two transportation system optimization scenarios, traffic signal control and dynamic lane assignment within junctions, and provides corresponding baseline methods for selection. You can execute it online and get the results of an hour of training based on the corresponding method.

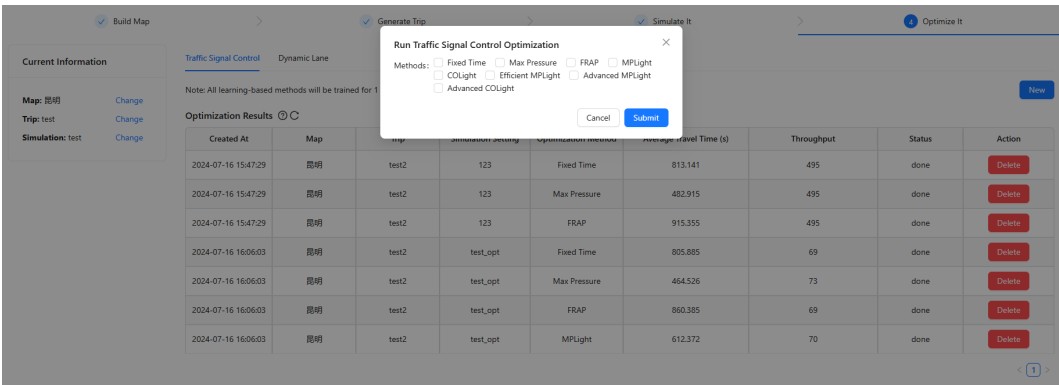

Figure A9: The optimization page of the web platform. (best view in color)

Moreover, we also write the documentation of our whole system including the simulator and the toolchain to guide researchers that are interested at coding to use the simulator. One of the documentation page is shown in Figure A10.

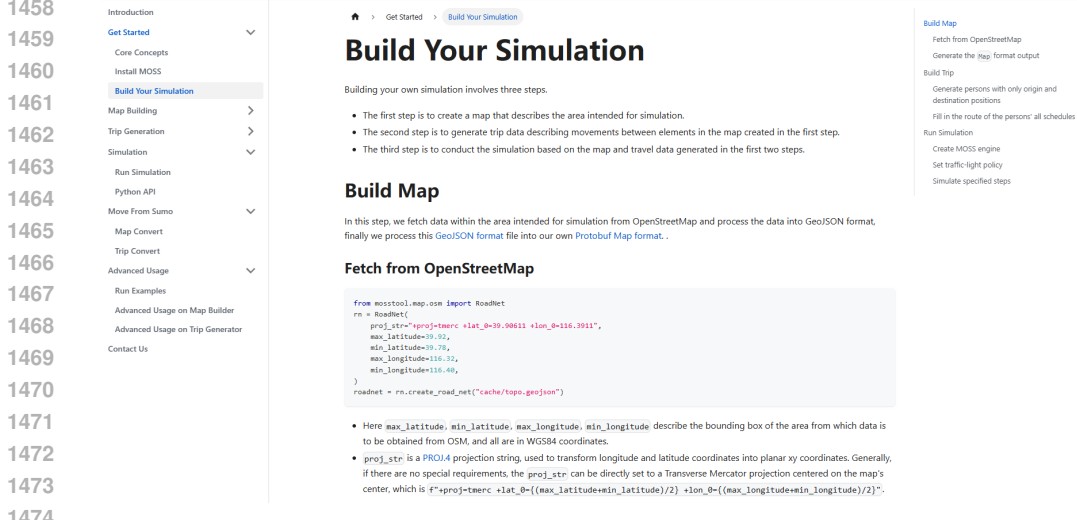

Figure A10: The documentation page of the web platform. (best view in color)

