# OpenReview forum: "A GPU-accelerated Large-scale Simulator for Transportation System Optimization Benchmarking"
_ICLR.cc/2025/Conference — Submitted to ICLR 2025_

### Official Review · Reviewer_YFJW · 2024-10-31

**Soundness:** 2
**Presentation:** 3
**Contribution:** 2
**Rating:** 5
**Confidence:** 4

**Summary:**

This paper proposes a microscopic simulator for large-scale urban traffic systems utilizing GPU acceleration. To overcome the computational limitations of existing simulators, the authors introduce GPU-based parallel processing, a two-phase parallel process, and a linked-list-based vehicle detection index. Through experiments simulating over 2.4 million vehicles, the proposed system achieved an approximately 88.92x performance improvement compared to existing systems. Furthermore, the authors present a framework that aims to integratively support five major traffic optimization scenarios: Traffic Signal Control, Dynamic Lane Assignment, Tidal Lane Control, Congestion Pricing, and Road Planning.

- Soundness (Score: 2 - Fair): The design and performance improvement of the GPU-accelerated simulator proposed in this research are technically convincing, and the experimental methodology is systematic. However, the simplifications, such as excluding intersection interactions and various traffic participants (e.g., pedestrians, bicycles, public transport), may significantly affect the realism of the simulation, and these aspects were not clearly validated. It is unclear whether prior research justified these exclusions. Although extensive benchmarking experiments were conducted, the lack of detailed analysis and interpretation of the results leaves the motivations behind the scenarios insufficiently explained.


- Presentation (Score: 3 - Good): The overall structure and explanations in the paper are clearly presented, particularly regarding system design and the use of GPU acceleration. However, the validation of each scenario within the same system is not clearly articulated. Additionally, it would be beneficial to provide a clear comparison of the proposed system architecture with existing studies, explaining the structural advancements and reasons for adopting this specific architecture.


- Contribution (Score: 2 - Fair): The proposed performance improvements through GPU acceleration and the integrated traffic simulator framework are meaningful contributions to the field. However, the lack of individual effectiveness analysis for the five scenarios and insufficient validation of the simulator's realism are notable shortcomings. The experiments with real-world data are limited, which poses a constraint on the practical contribution aspect.

**Strengths:**

Originality: The parallel design utilizing GPU features and the introduction of a linked-list-based vehicle detection index are promising approaches to overcoming the computational limitations of existing CPU-based traffic simulators.

Quality: The paper demonstrates a significant technical achievement by simulating over 2.4 million vehicles with GPU acceleration, achieving substantial performance improvement compared to CPU implementations. The system design and GPU usage are well-documented.

Clarity: The explanation of the system architecture is systematic.

Significance: To address urban traffic optimization, the authors propose an integrated simulation for five scenarios that were previously dealt with individually.

**Weaknesses:**

Scenario Validation and Motivation: It would be helpful to explain how the results from the five proposed scenarios provide insights for actual urban traffic planning or policy decisions. If empirical validation showing that similar results can be derived from simulating real-world policy changes were included, it would strengthen the motivation for policymakers to use the simulator. As a policymaker, I would want to validate planned changes beforehand with such a simulator, but the current motivation seems insufficient compared to the well-executed system implementation.

Lack of Realism: The simplifications, such as excluding intersection overlap, pedestrians, bicycles, and public transport, lack a clear basis for their effect on simulation outcomes. An analysis of the impact of these exclusions is necessary, including evidence from prior research or justifications for why their impact is minimal.

Insufficient Long-Term Scenario Evaluation: There is a lack of experiments evaluating how the proposed simulator functions under long-term urban traffic pattern changes. For example, reproducing outcomes of real-world urban policy changes using the simulator would help demonstrate its reliability.

Code Reproducibility Issues: Although the code was provided, the step-by-step explanations in the README.md were insufficient for successful reproduction. Specifically, two of the three provided source codes only included installation instructions without guidance on how to build or run scenarios. For the benchmark source code, even after locating it through references due to a missing GitHub link, the target version was not specified, preventing access to the necessary dataset for simulation. More detailed code instructions and thorough guidance on code operation are required, highlighting the lack of user-friendly considerations.

**Questions:**

Q1. Given the numerous experimental results presented, I felt that there was a lack of motivation for their inclusion. Could you provide additional explanations regarding the analysis and significance of these results?

Q2. There seem to be similar commercial services available (e.g., Aimsun). How does this work differentiate itself? There are also simulation games related to urban simulation, such as Cities: Skylines, which are known for their well-designed public transport systems, and the recent sequel has drawn significant attention. What are the commonalities and differences compared to such games? They support user-customized plugins for logging and debugging.

Q3. Could you explain the technical background that led to the design of the simulator's GPU-accelerated architecture, and how this design evolved from previous research efforts in the field?

Q4. Since the target is a simulator that is intended to be used alongside AI/ML-based technologies, which also use GPUs, resource constraints are expected if both the simulator and AI/ML technologies are to be used on servers that are not high-performance. Did you encounter this issue, and do you think it would be a significant problem?

** Additional Comments

There are inaccuracies in references.

- Unable to verify the original document. Only the citation is found. Please verify or update: HS Mahmassani. Dynamic traffic assignment and simulation for advanced network informatics (dynasmart). In the 2nd International Seminar on Urban Traffic Networks, 1992.

- The cited arXiv document version has been updated. Please verify or update: Alexander I Cowen-Rivers, Wenlong Lyu, Zhi Wang, Rasul Tutunov, Hao Jianye, Jun Wang, and Haitham Bou Ammar. Hebo: Heteroscedastic evolutionary bayesian optimisation. arXiv preprint arXiv:2012.03826, pp. 7, 2020.

- Duplicate reference entry. Please correct:
Qiang Wu, Ming Li, Jun Shen, Linyuan Lü, Bo Du, and Kecheng Zhang. Transformerlight: A novel sequence modeling based traffic signaling mechanism via gated transformer. Proceedings of the 29th ACM SIGKDD Conference on Knowledge Discovery and Data Mining, 2023a. URL: https://api.semanticscholar.org/CorpusID:260499801.
Qiang Wu, Mingyuan Li, Jun Shen, Linyuan Lü, Bo Du, and Ke Zhang. Transformerlight: A novel sequence modeling based traffic signaling mechanism via gated transformer. In Proceedings of the 29th ACM SIGKDD Conference on Knowledge Discovery and Data Mining, pp. 2639–2647, 2023b.

**Details Of Ethics Concerns:**

It is desirable to clarify the data sources used in the simulation.

- Privacy, Security, and Safety: The data used in the simulation includes urban traffic data, necessitating considerations for privacy and security.

- Potentially Harmful Insights, Methodologies, and Applications: The simulation results could have adverse effects if misused for real policy decisions, potentially disadvantaging specific regions or demographics. Ethical considerations are needed.

- Responsible Research Practice: For research involving urban data, it is essential to ensure transparency in legal/ethical approval procedures and the data collection and processing process.

---

> ### Author Response · Authors · 2024-11-24
> **Response to Reviewer YFJW**
>
> Thank you for your review.
>
> I will respond to your questions as follows:
>
> ## **To Weakness about Scenario Validation and Motivation**
>
> From our benchmarking results, optimization of signal control leads to the most significant results, followed by congestion pricing. There are two possible explanations for this. First, these two scenarios have received more attention and researchers have proposed some well-designed algorithms, while the other scenarios lack targeted optimization algorithms. One of the main contributions of our work is to propose these scenarios and provide benchmark datasets and simulators to boost the research works on optimization algorithms for these scenarios. Second, these approaches may also be more efficient means of managing and optimizing the transportation system. Our simulator can also be used to test the effects of combining multiple optimization methods. In addition, the development of research on transportation system optimization from computer simulation to real-world applications requires a lot of efforts, including but not limited to algorithmic innovations and real-world scenario validation. An important implication of our research work is to support interested researchers to make more innovations in transportation system optimization algorithms. We believe that only sufficient improvement in effectiveness will allow policymakers to start the long and tedious process of real-world scenario validation.
>
> ## **To Weakness about Lack of Realism**
>
> The simplifications, such as excluding intersection overlap, pedestrians, bicycles, and public transport, are consistent with established practices in microscopic traffic simulation, as seen in tools like SUMO, CityFlow, and CBLab. These choices are informed by prior research and are aimed at maintaining a balance between computational efficiency and realism.
> Although these simplifications do bring about a certain degree of realism reduction, in the results (Figure A3 in Appendix D) of the realism experiments given in this paper, we find that the road traffic status obtained from the simulations are similar to the real ones. We therefore believe that these simplifications are less likely to affect the results of the traffic simulation. On the other side, if reasonable traffic signal settings are provided in the real world (e.g., reasonable phase settings and clearance times), the percentage of intersection overlaps that severely impact vehicle movement in the intersections will occur will be small, and pedestrians and bicyclists will have a minimal impact on vehicle movement. This is the primary role of traffic signal. For public transportation, since this simulator mainly focuses on the movement of vehicles, buses can be considered as vehicles as well, while subways can be simply ignored. Overall, we consider the impact of these simplifications on realism to be within acceptable limits based on experimental results and logical reasoning based on common sense. This conclusion is also consistent with previous works.
>
> ## **To Weakness about Insufficient Long-Term Scenario Evaluation**
>
> The main contribution of this paper is to provide high-performance simulators capable of supporting large-scale traffic simulation simulations and to present several research-useful scenarios for optimizing transportation systems. The long-term effects of optimizing the urban transportation system using specific algorithms, such as changes in the travel patterns of residents, are not the subject of this paper. Such content can be investigated by interested researchers by combining the simulator with other techniques (e.g., historical data-based extrapolation, citizen decision-making simulation based on large model intelligences, etc.).
>
> ## **To Weakness about Code Reproducibility Issues**
>
> The dataset is now available at `https://iclr7274.obs.cn-north-4.myhuaweicloud.com/data.tar.gz` (8.57GB). For code, we provide an automated build process based on Github Action in a non-anonymized version and release Python packages on PyPI, and provide documentation with a zero-code wizard via a non-anonymized web service. Due to ICLR's double-blind policy, we are unable to provide the relevant content here. If the article is accepted, we will add the appropriate description in the Camera-ready version.
>
> (Page 1/2)

---

> > ### Comment · Reviewer_YFJW · 2024-11-26
> >
> > After reviewing the authors' rebuttal, I would maintain my current score with the following considerations:
> > 1. Scenario Validation and Motivation:
> > I understand that the main contribution of this research is providing a simulator framework to leverage the overall research community. The framework's potential impact on future research directions represents a significant contribution to the field.
> > 2. Regarding Realism in Simulation:
> > While you explain that the simplifications align with established practices in tools like SUMO, CityFlow, and CBLab, following precedent alone doesn't fully justify these simplifications. If there are prior research studies that specifically validate the impact of these simplifications with quantitative metrics, I would gladly reconsider my assessment of this aspect.
> > 3. Long-term Scenario Evaluation:
> > I understand that scenario settings are primarily within the user domain and separate from the main contribution. However, from a motivation perspective, demonstrating the simulator's potential for long-term urban planning scenarios could provide compelling use cases for potential users.
> > 4. Regarding Reproducibility:
> > While I understand ICLR's anonymity requirements present certain constraints, the installation documentation provided was insufficient for reviewers to properly evaluate the system. Even within anonymity constraints, the provided instructions should have been internally verified for usability. The late timing of dataset sharing has also limited our ability to conduct thorough validation, which is an important aspect of the review process.
> > (1/2 pages)

---

> ### Author Response · Authors · 2024-11-24
>
> ## **To Question 1**
>
> The primary purpose of including these experimental results in the main text is to demonstrate that our simulator is indeed capable of supporting the five proposed traffic simulation optimization scenarios. As a work that includes benchmarking, and in line with the writing style of previous articles of this kind, we are obliged to provide benchmarking results for the reference of subsequent researchers who conduct further research on the same datasets and simulators in order to minimize their meaningless time invested in repeating the experiments and to accelerate the forward progress of the research in the related fields.
>
> ## **To Question 2**
>
> - About Aimsum: We feel very sad because we don't own enough money and the city to get commercial services. So, based on the content of the introduction on Aimsum's homepage, we guess that its business services are implemented based on AI prediction methods, which may include some optimization algorithms as well. Our work provides a free algorithmic research environment for poor researchers like us to support their further innovations in the field of transportation system optimization.
> - About Cities Skylines: This game is a very good reproduction of urban planning, urban transportation (including congestion caused by poor signal management), and is a great primer for researchers optimizing transportation systems. However, in the game, you can't precisely control all the details, especially the speed of the traffic simulation. In addition, manually constructing a road network in the game similar to the real world, setting up travel routes for everyone, and developing tools for parsing game logs will be a huge pain for researchers. Therefore, specialized, high-performance, and code-friendly microscopic traffic simulators are a better choice for researchers in related fields.
>
> ## **To Question 3**
>
> Our research team has long focused on the application of AI technology to urban transportation problems. When we wished to optimize urban traffic systems using reinforcement learning algorithms, we found that existing simulators had such poor computational efficiency that simulations consumed unacceptable amounts of time when co-optimizing scenarios at multiple intersections. As a result, the idea of designing a large-scale high-performance microscopic traffic simulator arose. We learned the basic logic of running a microscopic traffic simulation from prior work in this area, including the construction of road networks, vehicle following models, vehicle lane changing models, and Python API interfaces. On the basis of these, we independently analyze the computational characteristics of microscopic traffic simulations and design corresponding solutions to achieve GPU computational acceleration.
>
> ## **To Question 4**
>
> If you only have 1 GPU, the AI algorithms will compete with the simulator for graphics memory. But if you have 2 GPUs, they can work on different GPUs and not interfere with each other. We don't think this is a serious problem, and we believe that most research teams can afford the extra investment of one GPU in the era of large language models.
>
> ## **To Additional Comments**
>
> Thank you very much for checking the references in detail. The DYNASMART reference can be found in the Google Scholar. We will update the citation of Hebo and delete the duplicated references.
>
> ## **To Ethics Concerns**
>
> The datasets we use are from open source data or previous works. The dataset for performance evaluation is from CBLab. The dataset for realism evaluation is from [1]. The datasets we use for benchmarks are built from OpenStreetMap and publicly available data represented by satellite imagery. No data on personal privacy is released in this work. We strongly agree with what you say about ethical considerations. If this paper is accepted, we will discuss it in the final version.
>
> [1] Yu, Fudan, et al. "City-scale vehicle trajectory data from traffic camera videos." Scientific data 10.1 (2023): 711.
>
> (Page 2/2)

---

> > ### Comment · Reviewer_YFJW · 2024-11-26
> >
> > Thank you for your responses which helped clarify several minor misunderstandings. I would appreciate additional clarification on the following points:
> > 1. Q1:
> > The work makes more sense when understood as a re-establishment of existing research with significant improvements.
> > 2. Q2, Regarding Commercial Solutions:
> > While I understand there may be practical limitations in direct comparisons with commercial solutions, focusing on implementation details and technical comparisons would be more constructive for academic discussion. Also, describing it as a solution "for poor researchers" seems inappropriate, especially given your framework's technical capabilities and potential commercial value.
> > 3. Q3, Regarding Framework Design:
> > From Figure 1, Figure A2 and your response, I understand you analyzed optimization bottlenecks, identified computational characteristics, and designed a GPU acceleration solution. I'm curious whether this represents a novel topology specific to traffic simulation, or if it's primarily a CUDA implementation of established frameworks. Could you clarify the key technical references or principles that influenced your architectural decisions?
> > 4. Q4, Regarding GPU Resources:
> > The clarification about GPU resource requirements helps understand the system's accessibility for research teams, which differs from my initial assumption about needing multiple GPUs.

---

### Official Review · Reviewer_KwpD · 2024-11-03

**Soundness:** 2
**Presentation:** 2
**Contribution:** 2
**Rating:** 5
**Confidence:** 2

**Summary:**

The paper presents a GPU-accelerated large-scale microscopic traffic simulator aimed at supporting optimization tasks within transportation systems. By leveraging GPU-based parallel computation, the simulator achieves high performance, simulating up to millions of vehicles at a significantly accelerated rate compared to traditional CPU-based simulators like CityFlow and CBLab. The simulator supports various transportation optimization scenarios, including traffic signal control, lane assignment, tidal lane control, congestion pricing, and road planning. The authors provide benchmark results across multiple cities to demonstrate the applicability and robustness of the simulator for different optimization algorithms.

**Strengths:**

## Originality
- The simulator is GPU-accelerated, which is helpful for traffic system simulation (while some GPU simulators are available for autonomous driving).

## Quality
- The paper provides benchmarks on various transportation optimization tasks, demonstrating practical applicability.

## Clarity
- The paper is well-organized and presents a clear problem statement, followed by the design choices and technical solutions implemented.

## Significance
- By supporting large-scale simulations at high speed, this simulator could enable faster and more frequent experimentation with AI-based optimization methods, including reinforcement learning.

**Weaknesses:**

- The paper lacks a comprehensive discussion of some relevant prior works. For example, recent works such as GPUDrive by Kazemkhani et al. (2024) and CityFlowER by Da et al. (2024) could provide essential insights and serve as useful points of comparison for evaluating the novelty and realism of the proposed simulator. For example, [1,3,4] are also a GPU simulators, are the techniques used for GPU acceleration in their papers the same as this paper? Is CityFlowER also as realistic as the proposed simulator?

[1] Kazemkhani, Saman, et al. "GPUDrive: Data-driven, multi-agent driving simulation at 1 million FPS." arXiv preprint arXiv:2408.01584 (2024).

[2] Da, Longchao, et al. "CityFlowER: An Efficient and Realistic Traffic Simulator with Embedded Machine Learning Models." Joint European Conference on Machine Learning and Knowledge Discovery in Databases. Cham: Springer Nature Switzerland, 2024.

[3] Saprykin, Aleksandr, Ndaona Chokani, and Reza S. Abhari. "GEMSim: A GPU-accelerated multi-modal mobility simulator for large-scale scenarios." Simulation Modelling Practice and Theory 94 (2019): 199-214.

[4] Jiang, Xuan, et al. "Large scale multi-GPU based parallel traffic simulation for accelerated traffic assignment and propagation." Transportation Research Part C: Emerging Technologies 169 (2024): 104873.

- The authors claim that individual vehicle simulation is compatible with the SIMD model, arguing that the vehicles are homogeneous and can be simulated in parallel. However, this assertion could be problematic as vehicle interactions, particularly in congestion, often exhibit dependencies on nearby vehicles. The paper would benefit from a clearer explanation of how dependencies are managed in parallel to ensure realism without sacrificing computational performance.

- In the Simulator Realism experiment, the paper mentions that the simulator achieves more realistic speeds compared to CityFlow. Given that CityFlowER now incorporates more realistic driving models, it would be valuable to benchmark the proposed simulator against it to provide a more comprehensive evaluation.

- When demonstrating scalability at large vehicle counts, the reported performance of CBLab appears to contradict with its original paper when simulating scenarios with over 1,000 vehicles, with CityFlow performing comparably or even outperforming in such instances. This discrepancy between claims and results needs further explanation on why CBLab is generally worse than CityFlow - is this because of your data is different from the CBLab? Can you provide more details about their experimental setup, including the specific datasets used and any differences from the original CBLab experiments? If so, can you use the similar dataset used in CBLab?  It would also be helpful to request a detailed comparison of their experimental setup with that of the original CBLab paper, including specifics on hardware configurations and simulation parameters.

-  The absence of scenario data within the provided code limits the reproducibility of results, particularly for scenarios such as traffic signal optimization and congestion pricing. Including this data would enhance transparency and reproducibility.

**Questions:**

1. Could the authors provide more justification for the assertion that vehicle simulations are “highly homogeneous” and compatible with the SIMD computational model? Specifically, how are dependencies between vehicles managed in congested scenarios?

2. Why does the simulator's performance appear to decline with scenarios of over 1,000 vehicles, as seen in the comparison with CityFlow? How does this relate to the scalability claims made in the paper?

3. Are there plans to include more detailed explanations of the simulation scenarios and the metrics used to evaluate them? This information would aid in validating the results for scenarios like congestion pricing and dynamic lane assignment.

4. Could the authors comment on the possibility of incorporating advanced driving models such as those used in CityFlowER, which have been shown to improve the realism of driving behavior?

---

> ### Author Response · Authors · 2024-11-24
>
> Really thank you for your review. We will answer your questions as follows:
>
> ## **To Weakness 1**
>
> Thank you very much for bringing to our attention the recent advancements in transportation simulation. Due to the earlier submission date of our manuscript, we were unable to incorporate these recent works from Arxiv or other publications. We appreciate your effort in highlighting these studies, which indeed show that this field is gaining significant attention, contrary to what reviewer 6yoX09 suggested.
>
> After thoroughly reviewing the manuscript and the open-source code of the relevant literature, here are our responses to your inquiries:
>
> 1. **GPUDrive**: This simulator focuses on autonomous driving scenarios, which significantly differs from our work in three key aspects:
>    - **Vehicle Movement**: In GPUDrive, vehicle movement is determined by input acceleration and angular velocity, with the road network serving merely as a benchmark for assessing the operational strategy's realism. In contrast, our simulator ensures vehicles move along lane lines, with car-following and lane-changing algorithms governing acceleration and lane-switching decisions.
>    - **Optimization Goals**: GPUDrive's external algorithms target the enhancement of individual vehicle autopilot algorithms. Our research, however, aims at optimizing city-wide traffic efficiency by modifying traffic infrastructure and operational strategies.
>    - **Simulation Scale**: GPUDrive supports hundreds of agents, whereas our simulator can handle over a million vehicles. Thus, our work belongs more to the domain of automated driving simulation rather than microscopic traffic simulation.
>
> 2. **CityFlowER**: This work seeks to enhance simulation realism by utilizing neural networks for car-following and lane-changing models. While this method holds potential for increased realism with sufficient real-world training data, such data is not reported in their study. Our focus is on achieving large-scale simulation with reasonable realism. According to the CityFlowER paper, it performs significantly worse than CityFlow and does not demonstrate higher realism, making it less significant in the context of related work. As for comparing the realism of our simulator with CityFlowER, the realism test data we have does not have detailed vehicle movement processes, only vehicle routes and road conditions, and therefore it is not possible to train the CityFlowER model to compare results. Nevertheless, we acknowledge CityFlowER's promise in microscopic traffic simulation.
>
> 3. **GEMSim**: This is a GPU-based mesoscopic traffic simulator, not a microscopic one. Mesoscopic simulators typically simplify vehicle behavior, akin to the waiting queue model used by MATSim, as mentioned in our paper. Our primary contribution is implementing large-scale parallel computational acceleration using a microscopic traffic simulation model to simulate traffic scenarios with high realism. This work can be included in the mesoscopic traffic simulation section of Related Works.
>
> 4. **Large scale multi-GPU based parallel traffic simulation for accelerated traffic assignment and propagation**: Although this paper claims to be a GPU-based microscopic traffic simulator, its model simplification approach — dividing roads into discrete 1-meter spaces — is uncommon. The paper does not address realism differences in simulation results. In contrast, our work achieves GPU acceleration without model simplification, ensuring realism.
>
> In summary, compared to these works, our work uniquely achieves large-scale microscopic traffic simulation without model simplification, maintaining the realism of the simulation results.
>
>
> ## **To Weakness 2 and Question 1**
>
> As what you said, how to manage the interaction between vehicles is an important issue to ensure realism. This is one of the core concerns as well as one of the main contributions of our work. In lines 226-228 of the original paper, we make it clear that a vehicle's sense range needs to include the vehicle in front of it in the current lane and the vehicle in front of it and behind it in the adjacent lane. To solve the problem, we implement SIMD-friendly vehicle sensing indexes and discuss the design in the lines 248-266. The simulation of individual vehicles is indeed homogenized when the sensing of the surrounding vehicles is taken into account (although there are times when the corresponding computations are not needed).
>
> (Page 1/2)

---

> ### Author Response · Authors · 2024-11-24
>
> ## **To Weakness 3 and Question 4**
>
> First and foremost, the Arxiv paper on CityFlowER does not provide any evaluation of realism; it only suggests that its methodology can replicate the driving behaviors observed in CityFlow and SUMO. Consequently, we do not consider CityFlowER to inherently offer a more realistic vehicle model. While CityFlowER indeed proposes a viable path towards achieving more realistic vehicle models, the ultimate realism is highly contingent upon the quality and scope of the training data used.
> Regrettably, the dataset we employed for evaluating realism lacked sufficient vehicle trajectory data necessary for effective training. This limitation precluded us from conducting a comprehensive evaluation. We acknowledge the importance of such comparisons and plan to incorporate advanced driving models in future work to enhance the realism of our simulator further.
>
> ## **To Weakness 4**
>
> For the performance evaluation, we directly utilized datasets from CBLab. Specifically, Roadnet-S corresponds to `eff_1e2inter` from CBLab's datasets, Roadnet-M corresponds to `Shanghai`, and Roadnet-L corresponds to `eff_1e6veh_1e4.5inter`.
> For the hardware configuration, we mentioned in the main text that the relevant information is provided in Appendix C. The CPU we use is Intel(R) Xeon(R) Platinum 8462Y CPU (64 threads).
> We do not comment on why the performance comparison between CBLab and CityFlow is inconsistent with CBLab's paper. First, the difference in performance between the two is not relevant to the innovation of this paper, and this paper is under no duty to reproduce the results reported by CBLab. Second, the computational performance of a software system is affected by a number of factors, such as the version of the compiler used, compilation options, the number of CPU cores, CPU cache, other loads running on the machine, etc. All we can do is to run baselines as fast as we can and report the results honestly.
>
> ## **To Weakness 5**
>
> We apologize for submitting the paper in an effort to find a suitable way to present the data without violating the double-blind principle. Now you can download the data at `https://iclr7274.obs.cn-north-4.myhuaweicloud.com/data.tar.gz` (8.57GB). If the paper is accepted, a great deal of content that is not publicly available due to the double-blind principle will be accessible, including the data structures we use, documentation, online services, and more.
>
> ## **To Question 2**
>
> For a software system, more computation means that it takes longer for the computer hardware to process, which reduces operational efficiency. As the number of vehicles in traffic simulation increases with the size of the road network, the increase in computational time is unavoidable. Our mention of “large-scale” implies that our proposed simulator is capable of performing computational tasks in city-scale scenarios with small time consumption, rather than implying that the scale of the simulation can be scaled infinitely without increasing the time spent.
>
> ## **To Question 3**
>
> The detailed explanation of the scenarios has been provided in Appendix F. The detailed explanation of the metrics has been provided in Section 3.3.
>
> (Page 2/2)

---

> > ### Comment · Reviewer_KwpD · 2024-11-25
> > **More related work discussion and comparison**
> >
> > Given the recent rebuttal from the authors, I tend to keep my current score with the following reasons:
> >
> > 1. Keeping Up with Related Work: I appreciate the acknowledgment of recent advancements, but honestly, the references I mentioned were all available before the ICLR deadline. It’s really important to stay on top of the latest research so the paper feels current and fully grounded in the existing work. Missing these makes the paper seem a bit out of touch with where the field is right now.
> >
> > 2. About the SIMD Model: The paper mentions the SIMD model as a key contribution, but to be fair, SIMD is a pretty standard technique in GPU-based simulations. The manuscript does not provide citations or a detailed discussion distinguishing the proposed SIMD implementation from existing techniques. This omission raises questions about whether this technique is novel or merely an application of existing methods. If so, it needs more explanation and citations to show how it stands out. Otherwise, it feels like it might just be a straightforward application of something that’s already well-known.
> >
> > 3. Since you’ve acknowledged the importance of realism comparisons and plan to enhance your simulator with advanced driving models in the future, one way to validate your current claims of realism could be to demonstrate how well your simulator replicates known simulation results. Showing lower errors or higher consistency with the same training data would strengthen your argument and provide a clearer benchmark for realism.
> >
> > 4. While I understand the claim that "the difference in performance between the two is not relevant to the innovation of this paper," it’s actually crucial for assessing the reliability of your experimental results. If the performance results don’t align with prior work, it raises questions about the faithfulness of the comparisons and the robustness of your evaluation. This directly impacts how your contribution is perceived, especially when realism and performance are core aspects of your claims.
> >
> >
> > A side comment after looking at other reviewers' comments: I’m not sure ICLR is the best place for this work. The paper tackles a really interesting problem, but it feels like it might get more attention and feedback at a conference that focuses on transportation or simulation research. ICLR is more about broader machine learning advancements, and this seems more niche, even though the problem itself is valuable.

---

> ### Author Response · Authors · 2024-11-25
>
> Thank you very much for your quick reply and we will post our further thoughts in response to your question.
>
> 1. There are no further comments, and we have fully explained the reasons for not including some of the work, and the way in which some of the work was included, in the previous comment.
> 2. SIMD itself is not the same thing as how to properly develop software using SIMD technology. In the case of the CPU, for example, this reviewer seems to be saying that all programs working on the CPU use the standard technology of the X86 instruction set, and therefore they are all simple applications of the X86 instruction set, and there is nothing innovative about them. We are very sad that the reviewer does not appear to have any background in computer expertise (e.g. Computer Architecture).
> 3. No more comment.
> 4. We sent an email to the original author of CBLab some time ago to discuss this and got no response. We are not responsible for the realism of the data reported in the CBLab paper, we are only responsible for the data reported in our paper.
>
> As to whether ICLR is the best place for this work, we believe that at least ICLR is the proper place for this work because this work is primarily aimed at researchers who wish to solve problems in urban transportation systems through reinforcement learning algorithms.

---

> > ### Comment · Reviewer_KwpD · 2024-11-25
> >
> > It’s sorry to see that the authors has been very defensive over my comments. My comment was “The manuscript does not provide citations or a detailed discussion distinguishing the proposed SIMD implementation from existing techniques.”
> >
> > I’ve been suggesting how to construct more convincing experiments and clearer writings on the novelty and sincerely hope the authors could consider them in the next draft.

---

### Official Review · Reviewer_oXrd · 2024-11-04

**Soundness:** 3
**Presentation:** 4
**Contribution:** 3
**Rating:** 6
**Confidence:** 4

**Summary:**

In this paper, the authors present a high-performance simulator for traffic system simulation and optimization, as well as the benchmark resulting for five scenarios with the simulator. The author clearly stresses the weakness of trending traffic system simulators and show through comparison that their proposed simulator can overcome the issues and outperform the SOTA one by running frequency and simulation realism. Overall, the presentation of the work is also clear and comprehensive, and tis work is good in its realm that from the comparison it does show the superiority in many aspects compared with existing simulators in the literature.

**Strengths:**

1.The proposed simulator is more efficient, realistic and capable compared to other candidates (sumo, cityflow, etc.). The proposed simulator is shown to be much faster than

2. The work has practical and application value, it can facilitate research and application in the relevant areas of traffic & transportation system. It is more versatile such that more component (traffic objects) are enabled in the simulator to be controllable. The authors also provide a few predefined evaluation metric APIs in the simulator to facilitate usage.

**Weaknesses:**

In Section 3, the authors described the design of each component of the simulator, however, it is now clear, which component is the main contribution of the design that improves the efficiency and performance of the traffic simulator. I would suggest the authorsadd a brief summary paragraph at the end of Section 3 that clearly states which specific components or techniques are the main technical contributions.

**Questions:**

Please see the limitation and clarify the technical contribution of the paper better. Some additional questions include:
1.Does the simulator allow the researcher to add uncertainty to the vehicle model or the control of traffic objects in order to validate the robustness of a method, and allow researchers to compare algorithms that reflect the simulation to reality gap? The authors please answer the following questions to clarify this point: 1) Describe any existing functionality for adding uncertainty/stochasticity, or 2) Discuss whether and how you plan to add such capabilities in future versions if not currently supported.


2. Will it be easy (or possible) to make the car-following model selectable (instead of sticking to IDM)? We suggest the authors to briefly discuss the modularity of the vehicle behavior models and what would be involved in adding support for multiple selectable car-following models. The answer should help clarify how flexible the simulator architecture is for future extensions.

---

> ### Author Response · Authors · 2024-11-13
> **Response to Reviewer oXrd**
>
> Thank you for your review.
> I will list the responses to your questions as follows:
>
> **To Weakness:**
> Overall, the two key designs presented in Section 3.1 are major contributors to improving efficiency and performance.
> In other words, both designs are indispensable. They are therefore of equal importance and should be considered as a whole.
> According to the idea of the paper, GPUs are the hardware base for enabling computational acceleration, and the task of software system design is to best adapt to the hardware.
> We identify two major difficulties (read/write conflict & vehicle sensing indexes) in the adaptation process and provide solutions (two-phase parallel process & linked-list based vehicle sensing indexes) that ultimately allow the GPU's performance to be fully utilized for acceleration.
>
> **To Question 1:**
> The answer is YES.
> Our simulator supports user-configurable inputs of kinematic and IDM model parameters for each vehicle, including maximum acceleration, general acceleration, maximum braking acceleration, general braking acceleration, headway, and so on.
> Users can build their own vehicle input data that they want to match the desired scenario, and we provide a tool chain (mosstool mentioned in Appendix A) to support such needs.
>
> **To Question 2:**
> It is easy to develop a new car-following model by editing a few codes.
> As shown in the `_CarFollow(Person& p, PersonNode& node, Lane* lane, float step_interval, uint& ahead_id)` function, we have implemented both the IDM model and the Krauss model.  (https://anonymous.4open.science/r/moss-AF45/src/entity/person/vehicle.cu: line 397)
> We hope that the open source code and the ICLR conference will attract more people to participate in the collaboration to provide more optional vehicle models.

---

> > ### Comment · Reviewer_oXrd · 2024-12-03
> > **Reply to the Rebuttal**
> >
> > Thanks you for the response. The authors have clarified my questions and concerns, I prefer to keep my score, since it is already slightly on the positive side for the work.

---

### Official Review · Reviewer_6yoX · 2024-11-09

**Soundness:** 3
**Presentation:** 3
**Contribution:** 2
**Rating:** 5
**Confidence:** 3

**Summary:**

This paper proposes a fast simulator for transportation system optimization.
The main contributions of this paper is a high-performance traffic flow simulation environment and the implementation of 5 transportation optimization problems. The new proposed simulation environment is GPU-accelerated and based on SIMD and programmed in CUDA. The performance of the proposed simulator is compared against some of the existing ones and has demonstrated superior performance on runtime. Different benchmark algorithms are demonstrated on the new simulator.

**Strengths:**

This work is a wonderful contribution for transportation system optimization, and I strongly believe many researchers can take advantage of this platform. The manuscript is very well structured and written well. The authors covers a comprehensive review of the existing simulations.

**Weaknesses:**

The scope of this work is too narrow, since there will only be a small subset of researchers that will use this in the learning community.

It will be nice to include an analysis of memory usage if possible.

**Questions:**

N/A

---

> ### Author Response · Authors · 2024-11-13
> **Response to Reviewer 6yoX**
>
> Thank you very much for taking your valuable time to review our paper.
>
> Our work, a large-scale microscopic traffic simulator, serves the fundamentals of transportation system optimization to support typical transportation system optimization scenarios.
> We believe that the efficiency of urban transportation systems is closely related to each individual, and that learning-based AI technologies have great potential to optimize the efficiency of transportation systems.
> However, as you say, there are fewer researchers focusing on this area right now.
> We believe this is because there is a lack of large-scale simulators and benchmarking code that can support a larger number of scenarios, leaving researchers unmotivated to investigate the application of learning methods to these problems.
> That is our aim in accomplishing such a non-incremental, pioneering and difficult work.
>
> Moreover, memory usage analysis will be added to the supplement as part of the performance evaluation.

---

> > ### Author Response · Authors · 2024-11-24
> >
> > ## Memory Usage
> >
> > | Simulator  | Roadnet-S | Roadnet-M | Roadnet-L |
> > | ---------- | --------- | --------- | --------- |
> > | SUMO       | 0.273GB    | 0.273GB    | 0.276GB   |
> > | CityFlow   | 2.42GB    | 3.11GB    | 14.62GB   |
> > | CBLab      | 1.70GB    | 5.30GB    | 99.76GB   |
> > | Ours (CPU) | 0.034GB   | 0.49GB    | 4.62GB    |
> > | Ours (GPU) | 0.44GB    | 3.94GB    | 16.82GB   |
> >
> > All tests used the maximum number of vehicles corresponding to the map dataset shown in Table A2 and recorded the maximum memory usage during the simulation run.
> > It is worth noting that the memory usage of our simulator includes both CPU memory and GPU memory, occupying two rows in the table.

---

### Author Response · Authors · 2024-11-24
**Welcome further discussion!**

We are pleased that all review comments have been responded to and we very much look forward to further communication with the reviewers and the interested public.

---

### Meta-Review · Area_Chair_Sp1H · 2024-12-29

**Metareview:**

The paper introduces a GPU-accelerated, open-source, large-scale microscopic simulator for transportation system simulation and optimization. The simulator achieves a significant computational acceleration and  supports a variety of transportation system optimization scenarios including traffic signal control, dynamic lane assignment, tidal lane control, congestion pricing, and road planning. The authors benchmark classical rule-based, reinforcement learning, and black-box optimization algorithms in four cities, demonstrating the simulator's usability for large-scale traffic system optimization.

The reviewers acknowledge the value of the simulator in enabling large-scale, efficient, and realistic transportation system simulations. While the simulator demonstrates high performance and versatility, the reviewers have raised significant concerns about its scope, novelty, the clarity of its technical contributions, the realism of the simulation, and the rigor of the experimental evaluation.

**Additional Comments On Reviewer Discussion:**

The authors' rebuttal addresses the reviewers' concerns by providing additional explanations and clarifications, but it does not fully resolve all of the issues raised. Some reviewers, maintain their concerns, particularly regarding realism, reproducibility, and novelty.

---

### Decision · Program_Chairs · 2025-01-22

Reject